Saltwater reduces potential $CO_2$ and $CH_4$ production in peat soils from a coastal freshwater
forested wetland
Kevan J. Minick[a]*, Bhaskar Mitra[b], Asko Noormets[b], John S. King[a]
*aDepartment of Forestry and Environmental Resources, North Carolina State University,*
*Raleigh, NC, 27695, USA*
*bDepartment of Ecosystem Science and Management, Texas A&M University, College Station,*
*TX, 77843, USA*
*Corresponding author: email kjminick@ncsu.edu; phone (919) 630-3307; fax NA*
Keywords: e*xtracellular enzyme activity, sea-level rise, methanogenesis, microbial biomass*
*carbon, carbon isotopes, ghost forest*
**Abstract** A major concern for coastal freshwater wetland function and health are the effects of
saltwater intrusion on greenhouse gas production from peat soils.  Coastal freshwater forested
wetlands are likely to experience increased hydroperiod with rising sea level, as well as saltwater
intrusion. These potential changes to wetland hydrology may also alter forested wetland structure
and lead to a transition from forest to shrub/marsh wetland ecosystems.  Loss of forested
wetlands is already evident by dying trees and dead standing trees ("ghost" forests) along the
Atlantic Coast of the US, which will result in significant alterations to plant carbon (C) inputs,
particularly that of coarse woody debris, to soils.  We investigated the effects of salinity and
wood C inputs on soils collected from a coastal freshwater forested wetland in North Carolina,
USA, and incubated in the laboratory with either freshwater or saltwater (2.5 or 5.0 ppt) and with
or without the additions of wood.  Saltwater additions at 2.5 ppt and 5.0 ppt reduced $CO_2$
production by 41 and 37 %, respectively, compared to freshwater.  Methane production was
reduced by 98 % (wood-free incubations) and by 75-87 % (wood-amended incubations) in
saltwater treatments compared to the freshwater plus wood treatment.  Additions of wood also
resulted in lower $CH_4$ production from the freshwater treatment and higher $CH_4$ production from
saltwater treatments compared to wood-free incubations.  The $\delta^{13}CH_4$-C isotopic signature
suggested that in wood-free incubations, $CH_4$ produced from the freshwater treatment originated
primarily from the acetoclastic pathway, while $CH_4$ produced from the saltwater treatments
originated primarily from the hydrogenotrophic pathway.  These results suggest that saltwater
intrusion into coastal freshwater forested wetlands will reduce $CH_4$ production, but long-term
changes in C dynamics will likely depend on how changes in wetland vegetation and microbial
function influence C cycling in peat soils.

**1 Introduction**

Sea-level rise (SLR) threatens coastal regions around the world. Significantly, the rate of SLR is not uniform around the globe, with the highest rate occurring along the Atlantic coast of North America between Cape Hatteras and Cape Cod due to factors including local currents, tides, and glacial isostatic rebound (Karegar et al., 2017; Sallenger et al., 2012). Along with economic and cultural impacts, health of coastal forested ecosystems are expected to be impacted by SLR (Langston et al., 2017; Kirwan and Gedan 2019). For instance, salinization of coastal freshwater wetlands will likely impact vegetation community dynamics and regeneration in low lying (< 1m) wetlands (Langston et al., 2017). Understanding how coastal wetland ecosystems respond to extreme events, long-term climate change and a rapidly rising sea is essential to developing the tools needed for sustainable management of natural resources, and the building of resilient communities and strong economies. Because it has more than 5,180 $km^2$ of coastal ecosystems and urban areas below 1 m elevation, the state of North Carolina is highly vulnerable to climate change and SLR and therefore saltwater intrusion (Riggs and Ames, 2008, Titus and Richman, 2001).

As sea level changes, coastal plant communities move accordingly up and down the continental shelf. In recent geologic time, sea level has risen about 3 m over the past ~2,500 years from sea level reconstructions adjacent to our study site (Kemp et al., 2011). The rate of SLR has varied greatly over that time, with periods of stability and change, and a geologically unprecedented acceleration in recent decades. The current distribution of coastal forested wetlands reflects the hydrologic equilibrium of the recent past climate, but the widespread mortality of such forests suggests that the rate of SLR is in a time of rapid change at a rate

potentially faster than the forest's capacity to move upslope, resulting in widespread death of
coastal freshwater forested wetlands (Kirwan and Gedan 2019).  Furthermore, dying coastal
forests will alter the quantity and quality of organic matter inputs to the soil as vegetation shifts
occur, as well as introduce a large pulse of woody debris into soils.  This has the potential to alter
C cycling processes responsible for storage of C in peat soils or loss of C as $CO_2$ and $CH_4$
(Winfrey and Zeikus, 1977).

Wetlands store more than 25% of global terrestrial soil C in deep soil organic matter

deposits due to their unique hydrology and biogeochemistry (Batjes, 1996; Bridgham et al.,
2006).  Carbon storage capacity is especially high in forested wetlands characterized by abundant
woody biomass, forest floors of *Spaghnum* spp., and deep organic soils.  Across the US
Southeast, soil organic C (SOC) in soils increases with proximity to the coast and is greatest in
coastal wetlands (Johnson and Kern, 2003).  Carbon densities are even higher in the formations
of organic soils (Histosols) that occur across the region, typically ranging from 687 to 940 t ha$^{-1}$,
but can be as high as 1,447 t ha$^{-1}$ (Johnson and Kern, 2003).  As noted, forested wetlands, which
historically have contributed to terrestrial C sequestration, are in serious decline and processes
leading to destabilization of accumulated soil C are not represented in broad-scale ecosystem and
land-surface models. The extent of changes in soil C cycling processes attributable to altered
hydroperiod, saltwater intrusion, and structural changes in vegetation in these ecosystems
remains unclear.

Saltwater intrusion, a direct result of SLR, into freshwater wetlands alters soil C cycling

processes (Ardón et al., 2016; Ardón et al., 2018), particularly that of methanogenesis (Baldwin
et al., 2006; Chambers et al., 2011; Dang et al., 2018; Marton et al., 2012), and microbial activity
(e.g., extracellular enzyme activity, Morrissey et al., 2014; Neubauer et al., 2013).  Saltwater
contains high concentrations of ions, particularly sulfate ($SO_4^{2-}$), which support high rates of
$SO_4^{2-}$ reduction compared to freshwater wetlands (Weston et al., 2011). Sulfate acts as a terminal
electron acceptor in anaerobic respiration of SOC, and $SO_4^{2-}$ reducers will typically increase in
abundance in response to saltwater intrusion and out-compete other anaerobic microorganisms,
particularly methanogens, for C (Bridgham et al. 2013; Dang et al., 2019; Winfrey and Zeikus,
1977).  The effect of $SO_4^{2-}$ on soil C cycling and competitive interactions with other anaerobic
microbial processes also appears dependent on the concentration of the ion (Chambers et al.,
2011).  Even within freshwater forested wetlands, hydrology and microtopography interact to
influence the amount of $SO_4^{2-}$ within soils experiencing different levels of saturation, and
therefore rates of $SO_4^{2-}$ reduction (Minick et al., 2019a).  A majority of saltwater intrusion
studies on soil C dynamics though have focused on tidal freshwater wetlands, whereas non-tidal
freshwater wetlands have received relatively little attention, partially due to their more confined
distribution across the landscape. Nonetheless, they occupy critical zones within the coastal
wetland ecosystem distribution and will be influenced by SLR differently than that of tidal
wetlands. Tidal wetlands may experience short-term pulses of saltwater with tidal movement of
water, while SLR effects on saltwater intrusion into non-tidal freshwater wetlands may result in
more long-term saltwater inundation.  This difference in saltwater inundation period may
influence rates of soil $CO_2$, $CH_4$ production, and microbial activity (Neubauer et al., 2013); and
therefore should be considered in light of the hydrologic properties of non-tidal wetlands.

Saltwater intrusion into freshwater systems may also influence the $CH_4$ production

pathways (Dang et al., 2019; Weston et al., 2011), as a result of saltwater-induced shifts in
methanogenic microbial communities (Baldwin et al., 2006; Chambers et al., 2011; Dang et al.,
2019).  Stable isotope analysis of $CO_2$ and $CH_4$ indicate that acetoclastic methanogenesis is the
major $CH_4$ producing pathway in freshwater wetlands (Angle et al., 2016), but the influence of
saltwater on the pathway of $CH_4$ formation in non-tidal freshwater forested wetlands has rarely
been studied, particularly through the lens of $CO_2$ and $CH_4$ stable C isotope analysis.  As $^{13}$C
isotopic analysis of $CH_4$ is non-destructive and is long-proven as a reliable indicator of the $CH_4$
production pathway (Whiticar et al., 1986), utilization of this analysis provides easily attainable
information on the effects of freshwater compared to saltwater on $CH_4$ production dynamics in
coastal wetland ecosystems experiencing SLR-induced changes in hydrology and vegetation.
Our goal in this study was to test whether saltwater additions alter the production of $CO_2$,
$CH_4$, and microbial activity from organic soils of a non-tidal temperate freshwater forested
wetland in coastal North Carolina, US, and whether effects differ in response to additions of
wood.  Although many studies have focused on salinity pulses in tidal freshwater wetlands, less
attention has been given to the effects of sustained saltwater intrusion on soil C dynamics.  We
expect saltwater intrusion due to SLR will be more persistent in non-tidal wetlands.  Therefore,
we investigated the effects of sustained saltwater inundation, using a laboratory microcosm
experiment, on greenhouse gas production and microbial activity (e.g., microbial biomass C and
extracellular enzyme activity).  Wood additions to microcosms were utilized to mimic the
potential large pulses of wood to peat soils as forest dieback occurs along the aquatic-terrestrial
fringes of the Atlantic Coast and these wetlands transition to shrub/marsh ecosystems (Kirwan
and Gedan 2019); thereby providing a large and widespread pulse of coarse woody debris to
wetland soils and potentially altering soil C cycling.

**2 Methods**

**2.1 Field Site Description**

The field site was located in the Alligator River National Wildlife Refuge (ARNWR) in Dare County, North Carolina (35°47'N, 75°54'W) (Figure 1). The ARNWR was established in 1984 and is characterized by a diverse assemblage of non-tidal pocosin wetland types (Allen et al., 2011). ARNWR has a network of roads and canals, but in general contains vast expanses of minimally disturbed forested- and shrub-wetlands. Thirteen plots were established in a 4 km$^2$ area in the middle of a bottomland hardwood forest surrounding a 35-meter eddy covariance flux tower (US-NC4 in the AmeriFlux database; Minick et al., 2019a). Of the 13 plots (7 m radius), four central plots were utilized for this study which have been more intensively measured for plant and soil properties and processes (Miao et al. 2013, Miao et al., 2017, Minick et al 2019a, 2019b, Mitra et al. 2019). Over-story plant species composition was predominantly composed of black gum (*Nyssa sylvatica*), swamp tupelo (*Nyssa biflora*), bald cypress (*Taxodium distichum*), with occasional red maple (*Acer rubrum*), sweet gum (*Liquidambar styraciflua*), white cedar (*Chamaecyparis thyoides*), and loblolly pine (*Pinus taeda*). The understory was predominantly fetterbush (*Lyonia lucida*), bitter gallberry (*Ilex albra*), red bay (*Persea borbonia*), and sweet bay (*Magnolia virginiana*). Mean air temperature and precipitation from climate records of an adjacent meteorological station (Manteo AP, NC, 35°55′N, 75°42′W, National Climatic Data Center) for the period of 2008 – 2018 was 17.0 ± 0.30 °C and 932 ± 38 mm, respectively. These wetlands are characterized by a hydroperiod that responds over short time scales and is driven primarily by variable precipitation patterns. Soils are classified as a Pungo series (very poorly drained dystic thermic typic Haplosaprist) with a deep, highly decomposed muck layer overlain by a shallow, less decomposed peat layer and underlain by highly reduced mineral sediments of

Pleistocene origin (Riggs, 1996).  Soils from the surface of hummocks have a pH of 4.2 ± 0.1, C
concentration of 49 ± 1.3 %, and a $\delta^{13}C$ value of -29.1 ± 0.29 ‰ (Minick et al. 2019b).  Ground
elevation is below < 1 m above sea level. Sea-level rise models of coastal NC show that
ARNWR will experience almost complete inundation by 2100, with attendant shifts in
ecosystem composition (DOD, 2010).

**2.2 Sample Collection**

Soil samples were collected on February 6, 2018, from surface organic soils by removing

seven 10 x 10 $cm^{-2}$ monoliths from hummocks to the depth of the root mat (approximately 6.3
cm) using a saw and a 10 x 10 $cm^{-2}$ PVC square.  The seven soil samples were composited by
plot and stored on ice for transport back to the laboratory.  In the laboratory, roots and large
organic matter were removed by hand and gently homogenized.  Soils samples were then stored
in the dark at 4°C for seven weeks before initiating the laboratory incubation.

Freshwater and saltwater for the experiment was collected from water bodies surrounding

the ARNWR on March 7, 2018 (Figure 1).  Freshwater was collected from Milltail Creek, which
runs Northwest from the center of ARNWR to Alligator River and drains our forested wetland
study site.  Freshwater salt concentration was 0 ppt.  Saltwater was collected from Roanoke
Sound to the east of ARNWR and had a salt concentration of 19 ppt (Figure 1).  Freshwater and
saltwater were mixed together to get the desired salt concentration for the saltwater treatments
(2.5 and 5.0 ppt).  These concentrations of saltwater were chosen due to the salinity levels in the
Croatan and Pamlico Sounds, which are adjacent to ARNWR (Figure 1).  Salinity in these waters
range from approximately 1 to 5 ppt (unpublished data). Prior to mixing, freshwater and
saltwater was filtered through a Whatman #2 filter (8 µm).  Neither saltwater nor freshwater
were sterile filtered, therefore microbial communities from each water source were mixed
together and added to the incubations. This could influence the response of soil microbes to the
various treatments, but also represents what would occur under future projections of SLR in this
region and the resulting mixing of freshwater and saltwater within the wetland.  Four water
samples of each freshwater and saltwater mixture were sent to the NCSU Environmental and
Agricultural Testing Service laboratory for analysis of total organic C (TOC), ammonium
($NH_4^+$), nitrate ($NO_3^-$), phosphate ($PO_4^-$), $SO_4^-$, calcium ($Ca^{2+}$), magnesium ($Mg^{2+}$), sodium
($Na^+$), potassium ($K^+$), and chlorine ($Cl^-$).  Analysis of TOC was made using a TOC analyzer
(Schimadzu Scientific Instruments, Durham, NC).  Analysis of $NH_4^+$, $NO_3^-$, and $PO_4^-$, was made
using Latchat Quikchem 8500 flow injection analysis system (Lachat Insturments, Milwaukee,
WI).  Sulfate and $Cl^-$ were measured on a Dionex ion chromatograph (Thermo Fisher Scientific,
Waltham, MA).  Finally, a Perkin Elmer 8000 inductively-coupled plasma-optical emission
spectrometer (Perkin Elmer, Waltham, MA) was used to analyze water samples for $Ca^{2+}$, $Mg^{2+}$,
$Na^+$, $K^+$, and $Cl^-$.

**2.3 Incubation Setup**

Incubation water treatments included: 1) soils incubated at 65 % water holding capacity
(WHC) (Dry); 2) soils incubated at 100% WHC with freshwater (0 ppt); 3) soils incubated at
100% WHC with a saltwater concentration of 2.5 ppt (2.5 ppt); and 4) soils incubated at 100%
WHC with a saltwater concentration of 5.0 ppt (5.0 ppt).  A subsample of each fresh soil (soils
stored at 4 ºC) was dried at 105°C to constant mass to determine gravimetric soil water content.
Approximately 150 – 200 g fresh soil (20 – 25 g dry weight) collected from each plot was
weighed into 1 L canning jars. For water addition estimates, WHC was calculated by placing a
subsample of fresh soil (approximately 2 g fresh weight) in a funnel with a Whatman #1 filter
and saturating with deionized $H_2O$ (d$H_2O$). The saturated sample was allowed to drain into a
conical flask for 2 h. After 2 h, the saturated soil was weighed, dried at 105 °C to constant mass,
and weighed again to determine WHC. It is important to note that the 100% WHC moisture
level resulted in soils being completely flooded (with either freshwater or saltwater) with water
covering the surface of the incubated soils, thereby allowing for the development of $CH_4$
producing conditions similar to that observed in the field for surface soils. After soil and water
additions, the remaining headspace was estimated for each individual incubation vessel
(approximately 750 mL) and used in the calculation of gas production rates. Following wood-
additions (see below), incubation vessels from each of the eight treatments were incubated in the
dark in the laboratory for 98 d at 20 – 23 ºC.
Two sets of incubations were set up with the above mentioned water treatments. We
added [13]C-depleted American sweetgum (*Liquidamber styraciflua*) wood to half the incubation
vessels (0.22 g wood per g soil) (wood-amended), while the other half were incubated without
wood (wood-free). Trees were grown at the Duke FACE site under elevated $CO_2$ concentrations
(200 ppm $CO_2$ above ambient) using natural gas derived $CO_2$ with a depleted [13]C signature
compared to that of the atmosphere (Feng et al., 2010; Schlesinger et al., 2006). The site was
established in 1983 after clear cut and burn (Kim et al., 2016). Trees were grown under elevated
$CO_2$ from 1994 to 2010 at which point they were harvested (Kim et al., 2016). Cookies were
removed from harvested trees, dried to a constant moisture level and stored at -20 °C until use.
The bark layer was removed and the outer six tree rings of multiple cookies were removed with a
chisel. Wood was then finely ground in a Wiley Mill (Thomas Scientific, Swedesboro, NJ,
USA) and analyzed for C content and $^{13}$C signature on a Picarro G2201-i Isotopic $CO_2/CH_4$
Analyzer outfitted with a Costech combustion module for solid sample analysis (Picarro Inc.,
Sunnyvale, CA USA). For $\delta^{13}$C analysis of solids (e.g., wood, microbial biomass extracts, soils),
certified solid standards were used to develop a standard curve from the expected and measured
$\delta^{13}$C values ($R^2 > 0.999$). These standards included USGS 40 (L-glutamic acid) ($\delta^{13}$C = -26.39
‰; USGS Reston Stable Isotope Laboratory, Reston, VA, USA), protein ($\delta^{13}$C = -26.98 ‰;
Elemental Microanalysis Ltd, Okehampton, UK), urea ($\delta^{13}$C = -48.63 ‰; Elemental
Microanalysis Ltd, Okehampton, UK), atropine ($\delta^{13}$C = -18.96 ‰; Costech Analytical
Technologies, Inc, Valencia, CA, USA), and acetanilide ($\delta^{13}$C = -28.10 ‰; Costech Analytical
Technologies, Inc, Valencia, CA, USA). For C concentration, atropine standards were weighed
out over a range of C concentrations that encompassed the expected C concentrations of the
unknown samples and within the measurement range of the instrument. A standard curve for C
concentration was also developed from the expected and measured C concentration of the
atropine standards ($R^2 > 0.99$). All unknown sample's C concentration and $\delta^{13}$C value were
adjusted using the linear equations derived from the appropriate standard curve. The $\delta^{13}$C values
were reported in parts per thousand (‰) relative to the Vienna Pee Dee Belemnite (VPDB)
standard. Wood had a C content of $45.6 \pm 0.21$ % and $\delta^{13}$C value of $-40.7 \pm 0.06$ ‰, which was
within the range of -42 to -39 ‰ measured on fresh pine needles and fine roots (Schlesinger et
al., 2006), and more depleted in $^{13}$C compared to that measured in hummock surface soils from
our site ($-29.1 \pm 0.29$ ‰; Minick et al. 2019b).

**2.4 $CO_2$ and $CH_4$ Sample Collection and Analysis**

Headspace gas samples were collected from incubation vessels 15 times over the course
of the 98 d incubation (days 1, 4, 8, 11, 15, 19, 25, 29, 29, 47, 56, 63, 70, 84, 98). Incubation lids
were loosened between measurements to allow for gas exchange with the ambient atmosphere.
Four blank incubations (empty jars; no soil, water, or wood) were set up and treated in the exact
same manner as incubations containing soil, water, and wood. Blanks were used to measure soil-
free $CO_2$ and $CH_4$ concentrations in incubations, which were always well below the detection
limit of the gas analyzer (described below). Prior to each measurement, incubation vessels were
removed from the dark, sealed tightly, and flushed at 20 psi for three minutes with $CO_2/CH_4$ free
zero air (Airgas, Radnor, PA, USA). Following flushing, incubation vessels were immediately
placed back in the dark (2-6 h over the first 39 days and 12-18 h over the remainder of the
incubation) before taking a gas sample for analysis. Approximately 300 mL of headspace gas
was removed using a 50 mL gas-tight syringe and transferred to an evacuated 0.5 L Tedlar gas
sampling bag (Restek, Bellefonte, PA, USA). Simultaneous analysis of $CO_2$ and $CH_4$
concentrations and $\delta^{13}C$ isotopic signature were conducted on a Picarro G2201-i Isotopic
$CO_2/CH_4$ Analyzer (Picarro Inc., Sunnyvale, CA USA). For $\delta^{13}C$ analysis of gases (e.g., $CO_2$
and $CH_4$), certified gas standards were used to develop a standard curve from the expected and
measured $\delta^{13}C$ values ($R^2 > 0.99$). The gas standards for $^{13}CO_2$ analysis included gas tanks
containing: 1) 372 ppm $CO_2$ with a $\delta^{13}C$ value of -11.0 ± 0.25 ‰ (Airgas, Inc., Radnor, PA); 2)
420 ppm $CO_2$ with a $\delta^{13}C$ value of -10.3 ± 0.18 ‰ (Airgas, Inc., Radnor, PA); 3) 768 ppm $CO_2$
with a $\delta^{13}C$ value of -29.5 ± 0.14 ‰ (Airgas, Inc., Radnor, PA); and 4) 3000 ppm $CO_2$ with a
$\delta^{13}C$ value of -34.4 ± 0.3 ‰ (Airgas, Inc., Radnor, PA). The gas standards for $^{13}CH_4$ analysis
included gas tanks containing: 1) 1.75 ppm $CH_4$ with a $\delta^{13}C$ value of -43.2 ± 0.07 ‰ (Airgas,
Inc., Radnor, PA); 2) 2.00 ppm $CH_4$ with a $\delta^{13}C$ value of -42.7 ± 0.20 ‰ (Airgas, Inc., Radnor,
PA); 3) 10.00 ppm $CH_4$ with a $\delta^{13}C$ value of -68.6 ± 1.00 ‰ (Airgas, Inc., Radnor, PA); and 4)
15.08 ppm $CH_4$ with a $\delta^{13}C$ value of  -29.5 ± 0.14 ‰ (Airgas, Inc., Radnor, PA).  For $CO_2$ and
$CH_4$ concentration, a concentrated gas standard (gas mix containing 4043 ppm $CO_2$ and $CH_4$)
(Airgas, Inc., Radnor, PA) was diluted with zero air gas, providing a range of $CO_2$ and $CH_4$
concentrations that encompassed the expected gas concentrations of the unknown samples.  A
standard curve for gas concentration was developed from the expected and measured gas
concentration of the diluted gas standards ($R^2 > 0.99$). All unknown gas sample $CO_2$ and $CH_4$
concentrations and $\delta^{13}C$ values were adjusted using the linear equations derived from the
appropriate standard curve.  The $\delta^{13}C$ values were reported in parts per thousand (‰) relative to
the Vienna Pee Dee Belemnite (VPDB) standard.  Production rates of $CO_2$-C and $CH_4$-C were
calculated as well as daily cumulative $CO_2$-C and $CH_4$-C production summed over the course of
the 98 d incubation.  Small subsamples (approximately 1.0 g dry weight) of soil were removed
periodically from each incubation vessel for extracellular enzyme analysis (see below).  Removal
of soil was accounted for in subsequent calculations of gas production rates.  Incubation vessel
water levels (mass basis) were checked and adjusted three times per week using either freshwater
or saltwater.

The proportion of wood-derived $CO_2$ at each sampling date was calculated using $^{13}CO_2$

data and the $^{13}C$ of depleted wood (-40.07) in a two pool flux model (Fry 2006), with the
depleted wood signature as one end-point and the $^{13}CO_2$ of wood-free incubations as the other
endpoint.

% C = (($\delta^{13}CO_{2\text{wood + soil}}$ - $\delta^{13}CO_{2\text{wood-free soil}}$) / ($\delta^{13}C_{\text{wood}}$ - $\delta^{13}CO_{2\text{wood-free soil}}$)) *100

Where $\delta^{13}CO_{2wood+ soil}$ is the $\delta^{13}C$ value of $CO_2$ produced from soils incubated with the
addition of $^{13}C$-depleted wood, $\delta^{13}C_{wood-free soil}$ is the $\delta^{13}C$ value of $CO_2$ produced from soils
incubated without the addition of $^{13}C$-depleted wood, and $\delta^{13}C_{wood}$ is the average $\delta^{13}C$ value of
the $^{13}C$-depleted wood.  Total wood-derived $CO_2$ was calculated using cumulative $CO_2$ produced
over the 98 d incubation and the average $^{13}CO_2$ across the whole incubation.

**2.5 Soil Characteristics**

Soil organic C concentration and $\delta^{13}C$ were analyzed on the four replicate soil samples
prior to the start of the incubation (initial soil samples) and on soils from each of the thirty
incubations following the 98 d incubation period.  The initial C analysis was performed on
samples removed prior to incubation.  Soils were finely ground in a Wiley Mill (Thomas
Scientific, Swedesboro, NJ, USA) prior to analysis on a Picarro G2201-i Isotopic $CO_2/CH_4$
Analyzer outfitted with a Costech combustion module for solid sample analysis (Picarro Inc.,
Sunnyvale, CA USA).  Carbon concentration and $^{13}C$ calibration standards were the same as
those described for the analysis of the $^{13}C$-depleted wood.
Soil pH and redox potential (Eh = mV) were measured in each incubation within one
hour following sampling of headspace gas. Soil pH was measured on the four replicate soil
samples immediately prior to the start of the incubation with a glass electrode in a 1:2 mixture
(by mass) of soil and distilled water (d$H_2O$).  Soil redox potential (Eh = mV) was measured
using a Martini ORP 57 ORP/ºC/ºF meter (Milwaukee Instruments, Inc., Rocky Mount, NC,
USA) .

## 2.6 Microbial Biomass Carbon and $\delta^{13}C$ Isotopic Signature


Microbial biomass C (MBC) was estimated on soils collected from incubations on day 1 (after 24 hour post-treatment incubation) and day 98 (following the end of the incubation). The chloroform fumigation extraction (CFE) method was adapted from Vance et al. (1987) in order to estimate MBC and $\delta^{13}C$. Briefly, one subsample of soil (approximately 0.5 g dry weight each) was placed in a 50 mL beaker in a vacuum desiccator to be fumigated. Another subsample was placed into an extraction bottle for immediate extraction in 0.5 M $K_2SO_4$ by shaking for 1 hr and subsequently filtering through Whatman #2 filter paper to remove soil particles. The samples in the desiccator were fumigated with ethanol-free chloroform ($CHCl_3$) and incubated under vacuum for 3 d. After the 3 d fumigation, samples were extracted similar to that of non-fumigated samples. Filtered 0.5 M $K_2SO_4$ extracts were dried at 60 °C in a ventilated drying oven and then ground to a fine powder with mortar and pestle before analysis of C concentration and $\delta^{13}C$ on a Picarro G2201-i Isotopic $CO_2/CH_4$ Analyzer outfitted with a Costech combustion module for solid sample analysis (Picarro Inc., Sunnyvale, CA USA). Carbon concentration and $^{13}C$ calibration standards were the same as those described for the analysis of the $^{13}C$-depleted wood. Microbial C biomass was determined using the following equation:


$$MBC = EC / k_{EC}$$


where the chloroform-labile pool (EC) is the difference between C in the fumigated and
non-fumigated extracts, and $k_{EC}$ (extractable portion of MBC after fumigation) is soil-specific
and estimated as 0.45 (Joergensen, 1996).
The $\delta^{13}C$ of MBC was estimated as the $\delta^{13}C$ of the C extracted from the fumigated soil
sample in excess of that extracted from the non-fumigated soil sample using the following
equation:

$$\delta^{13}C_{MBC} (\text{‰}) = ((\delta^{13}C_f \times C_f) - (\delta^{13}C_{nf} \times C_{nf}))/(C_f - C_{nf})$$

where $C_f$ and $C_{nf}$ is the concentration (mg kg$^{-1}$ soil) of C extracted from the fumigated
and non-fumigated soil samples, respectively, and $\delta^{13}C_f$ and $\delta^{13}C_{nf}$ is the $^{13}C$ natural abundance
(‰) of the fumigated and non-fumigated soil samples, respectively.

**2.5 Extracellular Enzyme Analysis**

The potential activity of five extracellular enzymes was quantified on soil samples and on
days 1, 8, 35, and 98 of the soil incubation.   The enzymes chosen for this experiment represent a
range of compounds in which they degrade, including fast and slow cycling C compounds, as
well as ones that target nitrogen, phosphorus, and sulfate containing compounds.  The Enzyme
Commission number (EC) is stated in parenthesis after each enzyme, which classifies them by
the chemical reaction catalyzed by each enzyme.  The specific enzymes measured were: β-
glucosidase (BG; EC: 3.2.1.21), xylosidase (XYL; EC 3.2.1.37), peroxidase (PER; EC: 1.11.1.7),
β-glucosaminidase (NAGase; EC: 3.2.1.30), alkaline phosphatase (AP; EC: 3.1.3.1), and
arylsulfatase (AS; EC: 3.1.6.1).  Carbon-degrading enzymes BG, XYL, and PER degrade sugar,
hemicellulose, and lignin, respectively, while the N-degrading enzyme, NAGase, degrades
chitin.  Enzymes AP and AS degrade phosphorus and $SO_4^{2-}$ containing compounds, respectively.
Substrates for all enzyme assays were dissolved in 50 mM, pH 5.0 acetate buffer solution for a
final concentration of 5 mM substrate.

Hydrolytic enzymes (BG, XYL, NAGase, AP, and AS) were measured using techniques

outlined in Sinsabaugh et al. (1993).  Approximately 0.8 g dry weight of soil sample was
suspended in 50 mL of a 50 mM, pH 5.0 acetate buffer solution and homogenized in a blender
for 1 min.  In a 2 mL centrifuge tube, a 0.9 mL aliquot of the soil-buffer suspension was
combined with 0.9 mL of the appropriate 5 mM p-nitrophenyl substrate solution for a total of
three analytical replicates. Additionally, duplicate background controls consisting of 0.9 mL
aliquot of soil-buffer suspension plus 0.9 mL of acetate buffer were analyzed, as well asfour
substrate controls consisting of 0.9 mL substrate solution plus 0.9 mL buffer.  The samples were
agitated for 2-5 hr.  Samples were then centrifuged at 8,160 g for 3 min.  Supernatant (1.5 mL)
was transferred to a 15 mL centrifuge tube containing 150 µL 1.0 M NaOH, followed by the
addition of 8.35 mL $dH_2O$.  The resulting mixture was vortexed and a subsample transferred to a
cuvette and the optical density at 410 nm was measured on a spectrophotometer (Beckman
Coulter DU 800 Spectrophotometer, Brea, CA, USA).

The oxidative enzyme (PER) was measured using techniques outlined in Sinsabaugh et

al. (1992).  PER is primarily involved in oxidation of phenolic compounds and depolymerization
of lignin.  The same general procedure for hydrolytic enzymes was followed utilizing a 5 mM L-
3,4-Dihydroxyphenylalanine (L-DOPA) (Sigma-Aldrich Co. LLC, St. Louis, MO, USA) solution
as the substrate plus the addition of 0.2 mL of 0.3% $H_2O_2$ to all sample replicates and substrate
controls.  After set up of analytical replicates and substrate and background controls, the samples
were agitated for 2-3 hr.  Samples were then centrifuged at 8,160 g for 3 min.  The resulting
supernatant turns an intense indigo color.  Supernatant (1.4 mL) was transferred directly to a
cuvette and the optical density at 460 nm was measured on a spectrophotometer.

For all enzymes, the mean absorbance of two background controls and four substrate

controls was subtracted from that of three analytical replicates and divided by the molar
efficiency (1.66/µmol), length of incubation (h), and soil dry weight.  Enzyme activity was
expressed as µmol substrate converted per g dry soil mass per hour ($\mu mol\ g^{-1}\ h^{-1}$).  Daily
cumulative enzyme activity was calculated and summed over the course of the 98 d incubation.

**2.6 Statistical Analysis**

Water chemistry, cumulative $CO_2$ production, cumulative $CH_4$ production, cumulative

enzyme activity, post-incubation SOC concentration and $\delta^{13}C$, and wood-derived and wood-
associated SOC, $CO_2$, and MBC were analyzed using a one-way ANOVA (PROC GLM
package).  Microbial biomass C, MBC $^{13}C$, pH, Eh, $\delta^{13}CO_2$, and $\delta^{13}CH_4$ were analyzed using
repeated-measures ANOVA (PROC MIXED package) with time (Time) as the repeated measure
and the incubation treatment as the fixed effect. All data for wood-free and wood-amended soils
were analyzed separately.  Raw data were natural log-transformed where necessary to establish
homogeneity of variance.  If significant main effects or interactions were identified in the one-
way or repeated-measures ANOVA ($P < 0.05$), then post-hoc comparison of least-squares means
was performed.  All statistical analyses were performed using SAS 9.4 software (SAS Institute,
Cary, NC, USA).

## 3 Results


### 3.1 Water and Soil Properties


Freshwater had higher concentrations of TOC compared to the saltwater treatments (Table 1). Concentration of $SO_4^{2-}$, $Cl^-$, $Na^+$, $Ca^{2+}$, $Mg^{2+}$, and $K^+$ were higher in saltwater treatments compared to freshwater and were approximately twice as high in the 5.0 ppt saltwater treatment compared to 2.5 ppt saltwater (Table 1).

Initial (pre-incubation) SOC concentration was $490 \pm 27$ g kg$^{-1}$ with a $\delta^{13}C$ value of -28.5 $\pm$ 0.32 ‰. After 98 d of incubation, SOC concentration in wood-free incubations was lower in the 5.0 ppt saltwater treatment, although no difference in soil $\delta^{13}C$ was found between treatments (Table 2). For wood-amended incubations, post-incubation SOC concentration was lower in the 5.0 ppt saltwater treatment compared to the dry and freshwater treatment (Table 2). Overall, the $\delta^{13}C$ of wood-free (-29.5 $\pm$ 0.08 ‰) and wood-amended soils (-30.5 $\pm$ 0.12 ‰) after 98 days of incubation were significantly different (F = 49.6; $P < 0.0001$).

Soil pH was significantly lower in the saltwater treatments in both wood-free and wood-amended soils compared to the dry and freshwater treatments (Table 3; Figure 2A-B). After an initial drop of pH in saltwater treatments (wood-free and wood-amended) to between 3.2 and 3.4 pH, pH steadily climbed back up to between 3.8 and 4.2 pH (Figure 2A-B). In wood-free soils, differences in soil Eh between treatments was variable over time, with both the 5.0 ppt saltwater treatment and the freshwater treatment having the lowest redox potential at different time points throughout the incubation (Table 3; Figure 2C), but fell below -124 mV on average. In wood-

amended soils, Eh dropped quickly to between -200 and -400 mV over the first 30 days for
saltwater incubated soils (Table 3; Figure 2D), before rising to between -100 to 0 mV for the rest
of the incubation period. In freshwater incubated soils, Eh rose quickly back to between -50 to 50
mV by day 15 and remained at this level for the rest of the incubation period, while saltwater
treatments had significantly lower Eh between days 8 and 25.

**3.2 $CO_2$, $CH_4$, $\delta^{13}CO_2$-C, and $\delta^{13}CH_4$-C**

In wood-free incubations, cumulative $CO_2$ production was not different between the dry
and freshwater treatments, but was higher than that produced from saltwater treatments (Table 4;
Figure 3A).  Cumulative $CO_2$ produced from wood-amended soils was highest in the dry
treatment compared to all other treatments (Table 4; Figure 3B). Wood-derived $CO_2$ (calculated
as the difference between cumulative $CO_2$ produced from wood-amended and wood-free
incubations) was highest in the dry treatment (Table 4; Figure 3C).  This finding was also
confirmed by calculating cumulative wood-derived C using the $^{13}C$ two-pool mixing model, with
the highest proportion found in the dry treatment ($54 \pm 4.6$ %) compared to soils incubated with
freshwater ($42 \pm 1.7$ %), 2.5 ppt saltwater ($37 \pm 1.0$ %), and 5.0 ppt saltwater ($38 \pm 1.5$ %) (F =
10.1; $P = 0.001$).
Cumulative $CH_4$ production was highest in the freshwater treatment compared to the
saltwater treatments in both wood-free and wood-amended incubations (Table 4; Figure 3D-E).
The difference between cumulative $CH_4$ produced from wood-amended and wood-free
incubations was lower (and exhibited a negative response to wood additions) in the freshwater
treatment compared to both saltwater treatments (Table 3; Figure 3F), which both had a slight
positive response to wood additions.

The $CO_2$:$CH_4$ ratio, in wood-free incubations, was calculated only for soils incubated

under saturated conditions with freshwater or saltwater. The $CO_2$:$CH_4$ ratio, in wood-free
incubations, was highest in freshwater (6 ± 3.4), compared to the 2.5 ppt saltwater (136 ± 33.9)
and 5.0 ppt saltwater (102 ± 30.3) (F = 24.8; $P$ = 0.0002). The $CO_2$:$CH_4$ ratio, in wood-amended
incubations, was highest in freshwater (9 ± 0.8), compared to the 2.5 ppt saltwater (53 ± 20.3)
and 5.0 ppt saltwater (107 ± 37.7) (F = 9.2; $P$ = 0.007).

The $\delta^{13}CO_2$-C and wood-derived $CO_2$ (estimated by $^{13}C$ two-pool mixing model)

exhibited a time by treatment interaction for both wood-free and wood-amended incubations
(Table 3; Figure 4A-B).  In general, $\delta^{13}CO_2$-C in wood-free and wood-amended incubations was
depleted in the dry treatment (and remained steady throughout the incubation period) compared
to all other treatments, especially after day 15.  The proportion of wood-derived $CO_2$ was
initially higher in freshwater and saltwater treatments (after day 1) but gradually dropped over
the course of the incubation, while the proportion of wood-derived $CO_2$ from the dry treatment
dropped quickly after the first sampling date (day 1) and remained steady (approximately 50-60
%) for the remainder of the incubation period (Figure 4C).

The $\delta^{13}CH_4$-C (Table 3; Figure 5) exhibited a treatment and time effect (Table 3; Figure

5A-B), but only for wood-free incubations.  For wood-free incubations, average $^{13}CH_4$-C across
the course of the incubation was enriched in the freshwater treatment (-67.8 ± 2.4 ‰) compared
to the 2.5 ppt (-80.1 ± 2.4 ‰) and 5.0 ppt (-82.3 ± 2.0 ‰) saltwater treatments (Figure 5C).  No
difference in the $\delta^{13}CH_4$-C was found in wood-amended incubations (Figure 4b, d), which
ranged from between -78 to -75 ‰ for all treatments.

## 3.3 Microbial Biomass Carbon and Extracellular Enzyme Activity

Initially, MBC was lowest in the dry treatment of wood-free incubations and in the 5 ppt treatment of wood-amended incubations (Table 3; Table 5). Following the 98 day incubation, MBC was highest in the dry treatment of wood-free incubations, with no differences between the other treatments. In wood-amended incubations, final MBC was also highest in the dry treatment compared to both saltwater treatments. Initial $\delta^{13}C$ of MBC did not differ between treatments in either the wood-free or wood amended soils (Table 3; Table 5). After the 98 day incubation, $^{13}C$ of MBC in the wood-free treatments was depleted in the freshwater treatment and enriched in the 5.0 ppt saltwater treatment. In wood-amended incubations, $^{13}C$ of MBC was depleted in the dry treatment and enriched in the freshwater and 5.0 ppt saltwater treatments. Furthermore, the proportion of wood-derived MBC (as estimated by $^{13}C$ mixing model calculations) was highest in the dry treatment (31 %) and the 2.5 ppt saltwater treatment (21%) compared to the freshwater treatment (4%) (Table 5).

In wood-free incubations, activity of BG and NAGase was higher, while PER was lower, in the dry treatment compared to the saltwater treatments (Table 4; Table 5). Activity of AS was higher in the dry and freshwater treatments compared to saltwater treatments, in both wood-free and wood-amended incubations. In wood-amended incubations, BG and NAGase were highest in the dry treatment compared to the saltwater treatments. In the freshwater treatment, wood addition reduced activity of BG and NAGase compared to wood-free incubations (Figure 6A-B), but enhanced PER activity (Figure 6C). Wood addition also reduced AS and P activity across all treatments compared to wood-free incubations (Figure 6D-E).


**4 Discussion**


As forests within the lower coastal plain physiographic region of the southeastern US

continue to experience increasing stresses from SLR, changes in microbial C cycling processes
should be expected.  Our results, combined with other field and lab experiments, confirm that
saltwater intrusion into coastal freshwater forested wetlands can result in reductions in $CO_2$ and
$CH_4$ production (Ardón et al., 2016; Ardón et al.,2018), but this may be balanced by long- and
short-term effects of saltwater intrusion on these C cycling processes (Weston et al., 2011), as
well as changes in C inputs due to forest-to-marsh transition.  Further, wood additions to these
wetland soils may reduce $CH_4$ production under freshwater conditions compared to the absence
wood additions (Figure 3C and 3F), but slightly enhance $CH_4$ production under saltwater
conditions.  Our results also demonstrate that substantial quantities of $CH_4$ can be produced from
freshwater wetland soils with redox potential between -100 to 100 mV, which may be related to
the specific pathway of $CH_4$ production (acetoclastic versus hydrogenotrophic) (Angle et al.,
2016), and challenges the widespread assumption that methanogenesis only occurs at very low
redox potentials.  Changes in the water table depth at the ARNWR is driven primarily by
precipitation patterns (Minick et al., 2019a), resulting in the influx of oxygenated waters.
Periodic *in situ* measurements of redox potential at the ARNWR indicate that standing water is
relatively aerated (Eh = 175 - 260 mV), while surface soils of hummocks when not submerged
are more aerated (Eh = 320 mV) than submerged hollow surface soils (Eh = 100 to 150 mV) and
deeper organic soils (20 – 40 cm depth; Eh = 50 to 90 mV) (unpublished data).  Furthermore, our
results indicate that additions of new C to soils as wood may result in short-term reductions in
redox potential as anaerobic processes are enhanced due to the added C substrate and terminal
electron acceptors are quickly reduced.  As SLR continues to rise over the next century, more
persistent saltwater intrusion may occur as rising brackish waters mix with non-tidal freshwater
systems having important implications for both above- and below-ground C cycling dynamics.
Although our study only looked at these effects in a controlled laboratory experiment, these data
provide a baseline understanding of potential changes in C cycling dynamics in these wetlands
due to SLR.

Saltwater additions decreased $CO_2$ production compared to freshwater in the wood-free

soils, although post-incubation MBC and extracellular enzyme activity (e.g., BG, NAGase, and
AP) were not different between these treatments. This has been found in other pocosin wetland
soils on the coast of North Carolina (Ardón et al. 2018). Variable effects of salinity (and/or $SO_4^{2-}$
additions) have been found on soil respiration, with some studies showing an increase (Marton et
al., 2012; Weston et al., 2011), a decrease (Lozanovska et al. 2016; Servais et al. 2019), or no
change (Baldwin et al., 2006).  Krauss et al. (2012) found that permanently flooded saltwater
treatments (expected in non-tidal wetlands) in a simulated coastal swamp mesocosm reduced soil
respiration, whereas saltwater pulses (expected in tidal wetlands) had a variable effect on soil
respiration.  Alternatively, $CO_2$ production was not reduced in the saltwater compared to
freshwater treatments in wood-amended soils, while post-incubation MBC was lower in the
saltwater compared to freshwater, which suggests a shift in microbial carbon use efficiency.

Methane production was higher in the freshwater compared to saltwater treatments in

both wood-amended and wood-free incubations.  Numerous others studies have found that
saltwater reduces $CH_4$ fluxes compared to freshwater, both within the field and laboratory.
Reduced $CH_4$ production from saltwater treated soils primarily results from the availability of
more energetically favorable terminal electron acceptors (primarily $SO_4^{2-}$), which leads to the
competitive suppression of methanogenic microbial communities by $SO_4^{2-}$ reducing communities
(Bridgham et al., 2013; Chambers et al., 2011; Winfrey and Zeikus, 1977), as methanogens and
$SO_4^{2-}$ reducers compete for acetate and electrons (Le Mer and Roger, 2001).  Dang et al. (2019)
did find partial recovery over time of the methanogenic community following saltwater
inundation to freshwater soil cores, but interestingly this community resembled that of microbes
performing hydrogenotrophic methanogenesis and not acetoclastic methanogenesis.  Activity of
arylsulfatase was also lower in saltwater amended soils.  This also indicates a functional change
in the microbial community, as microbes in the saltwater treatment are utilizing the readily
available $SO_4^{2-}$ pool, while microbes in the freshwater and dry treatments are still actively
producing $SO_4^{2-}$-liberating enzymes to support their metabolic activities.  Findings by Baldwin et
al. (2006) support the effects of saltwater on changing the microbial community structure as
well, in which reductions in $CH_4$ production in NaCl treated freshwater sediments were
accompanied by a reduction in archaeal (methanogens) microbial population, establishing a link
between shifting microbial populations and changing $CH_4$ flux rates due to saltwater intrusion.

Changes in the $CH_4$ production due to saltwater additions appear to be related to the

dominant $CH_4$ producing pathway.  The $^{13}CH_4$ isotopic signature in wood-free freshwater
incubated soils indicated that acetoclastic methanogenesis was the dominant $CH_4$ producing
pathway, while hydrogenotrophic methanogenesis dominated in the saltwater treatments.
Acetoclastic methanogenesis produces isotopically enriched $CH_4$ compared to that of the
hydrogenotrophic methanogenesis (Chasar et al., 2000; Conrad et al. 2010; Krohn et al. 2017;
Sugimoto and Wada, 1993; Whiticar et al., 1986; Whiticar 1999). The differences in C
discrimination between the two pathways is greater for the hydrogenotrophic compared to the

573 acetoclastic pathway, resulting in more depleted (-110 to -60 ‰) and more enriched (-60 ‰ to -

574 50 ‰) $^{13}CH_4$, respectively.  This has been confirmed in field and laboratory experiments (Conrad

575 et al. 2010; Krohn et al. 2017; Krzycki et al., 1987; Sugimoto and Wada, 1993; Whiticar et al.,

576 1986; Whiticar, 1999).  Baldwin et al. (2006) also found that saltwater additions promoted the

577 hydrogenotrophic methanogenic pathway.  Further, recent studies have found that saltwater

578 additions to soils result in a shift in the relative abundance of hydrogenotrophic methanogens

579 (Chambers et al. 2011; Dang et al 2019), supporting the idea that saltwater may alter not only the

580 production of $CH_4$ but also the pathway of methane production.

581  Changes in freshwater and saltwater hydrology due to rising seas is leading to dramatic

582 shifts in the dominant plant communities within the ARNWR and across the southeastern US

583 (Connor et al., 1997; DOD, 2010; Langston et al., 2017; Kirwan and Gedan 2019). This has the

584 potential to alter the soil C balance due to introduction of large amounts of coarse woody debris

585 as trees die.  In our laboratory experiment, additions of wood resulted in changes in both $CO_2$

586 and $CH_4$ production, but the direction of change depended on if soils were incubated with

587 freshwater or saltwater.  Wood additions increased $CO_2$ production compared to wood-free soils,

588 except in the freshwater treatment.  This was particularly evident in the dry treatment where

589 wood additions increased $CO_2$ production by approximately 32 %.  For the dry treatment, wood-

590 amended soils had the highest MBC and NAGase activity as microbes were likely immobilizing

591 more N to support metabolic activities in the presence of added C (Fisk et al., 2015; Minick et

592 al., 2017). Higher respiration with wood additions in the saltwater treatments likely resulted from

593 enhanced metabolic activity of $SO_4^{2-}$ reducing microbes in the presence of an added C source.

594 On the other hand, wood additions resulted in a decline in $CH_4$ production from the freshwater

595 treatment, while slightly enhancing $CH_4$ production from the saltwater treatments.  Wood

additions also resulted in much lower redox potential, particularly in the saltwater treatments,
and coupled with $^{13}CH_4$ stable isotope composition may have driven the higher levels of $CH_4$
production (via hydrogenotrophic methanogenesis) in the wood plus saltwater treatments.  The
suppression of $CH_4$ production by wood additions in the freshwater treatment was somewhat
surprising given the positive effects of C additions on $CH_4$ production recently found in
freshwater sediments (West et al. 2012), but likely resulted from enhancement of other, more
energetically favorable redox reactions with the addition of a C source (e.g., wood). Furthermore,
wood additions to freshwater incubations resulted in a decrease in MBC and activity of BG and
NAGase enzymes compared to wood-free incubations and an increase in PER activity.  This
suggests that the microbial communities have altered their functional capacity in response to
wood additions when exposed to freshwater.  The $CO_2{:}CH_4$ ratio further indicated that, in
freshwater, $CH_4$ production was quite high in relation to $CO_2$ production.  This ratio was
significantly higher for saltwater treatments as $CH_4$ production dropped drastically compared to
freshwater.  In wood-free incubations, the $CO_2{:}CH_4$ trend between freshwater and saltwater
treatments was parabolic, but was linear upward in wood-amended soils. This suggests that
interactions between saltwater concentration and coarse woody debris (in the form of dead and
dying trees; Kirwan and Gedan 2019) may be important to understand when determining effects
of saltwater intrusion on greenhouse gas production in freshwater forested wetlands.

Findings from this study indicate that substantial changes in the greenhouse gas

production and microbial activity are possible due to saltwater intrusion into freshwater wetland
ecosystems but that the availability of C in the form of dead wood (as forests transition to marsh)
may alter the magnitude of this effect.  At ARNWR and similar coastal freshwater forested
wetlands, saltwater intrusion may reduce both $CO_2$ and $CH_4$ emissions from soils to the
atmosphere.  Sea-level rise will likely lead to dramatic and visually striking changes in
vegetation, particularly transitioning forested wetlands into shrub or marsh wetlands (Kirwan and
Gedan 2019), which has resulted in the widespread occurrence of "ghost" forests along the
Atlantic coast (Kirwan and Gedan 2019).  As forested wetlands are lost, dead trees could provide
a significant source of C to already C-rich peat soils, with the potential to alter $CO_2$ and $CH_4$
production.  The long-term effect of forest-to-marsh transition on ecosystem C storage will likely
depend on the balance between dead wood inputs and effects of SLR and vegetation change on
future C inputs and soil microbial C cycling processes.  Future work should include investigation
of these C cycling and microbial processes at the field-scale and expand to a wider range of non-
tidal wetlands within the southeastern US region.

**Author contribution**

All authors contributed to the conception and design of the study. KM wrote the first draft of the
manuscript. KM collected the samples from the field and performed laboratory analysis. All
authors contributed to manuscript revision and approved the submitted version.

**Competing Interest**

The authors declare that they have no conflict of interest.

**Acknowledgements**

We thank numerous undergraduate researchers for their invaluable help collecting samples from
the field and analyzing samples in the laboratory.  We also thank two reviewers for their
comments, which significantly improved the manuscript.  Primary support was provided by
USDA NIFA (Multi-agency A.5 Carbon Cycle Science Program) award 2014-67003-
22068.  Additional support was provided by DOE NICCR award 08-SC-NICCR-1072, the
USDA Forest Service Eastern Forest Environmental Threat Assessment Center award 13-JV-
11330110-081, and Carolinas Integrated Sciences and Assessments award 2013-0190/13-
2322.  The USFWS Alligator River National Wildlife Refuge provided helpful scientific
discussions, the forested wetland research site, and valuable in–kind support.

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

**Tables and Figures**

Table 1. Total organic C (TOC) and ion concentrations (mg L$^{-1}$) in freshwater (0 ppt), 2.5 ppt saltwater, and 5.0 ppt saltwater.
Standard errors of the mean are in parenthesis (n=4). Values with different superscript lowercase letters are significantly different ($P <$
0.05).

| Treatment | TOC | $SO_4^{2-}$ | $Cl^-$ | $Na^+$ | $NH_4^+$ | $NO_3^-$ | $PO_4^{3-}$ | $Ca^{2+}$ | $Mg^{2+}$ | $K^+$ |
|---|---|---|---|---|---|---|---|---|---|---|
| 0 ppt | 44 (0.3)$^a$ | 1 (0.1)$^a$ | 17 (0.2)$^a$ | 8 (0.1)$^a$ | 0.00 (0.000)$^a$ | 0.00 (0.000)$^a$ | 0.00 (0.000)$^a$ | 1 (0.0)$^a$ | 1 (0.0)$^a$ | 0.2 (0.0)$^a$ |
| 2.5 ppt | 40 (0.7)$^b$ | 162 (1.3)$^b$ | 1391 (42.8)$^b$ | 538 (19.2)$^b$ | 0.06 (0.004)$^b$ | 0.06 (0.000)$^a$ | 0.01 (0.000)$^a$ | 23 (0.3)$^b$ | 64 (2.6)$^b$ | 19 (0.3)$^b$ |
| 5.0 ppt | 38 (0.1)$^b$ | 319 (6.5)$^c$ | 2695 (22.6)$^c$ | 1039 (15.9)$^c$ | 0.07 (0.004)$^b$ | 0.07 (0.004)$^a$ | 0.01 (0.000)$^b$ | 44 (1.0)$^c$ | 125 (2.1)$^c$ | 36 (0.4)$^c$ |












Table 2. Post-incubation soil organic C (SOC) concentration (g kg$^{-1}$), SOC $\delta^{13}$C (‰), and wood-derived SOC (%) (estimated from $^{13}$C
two pool mixing model) for soil samples collected from the field and incubated for 98 d in the laboratory under dry conditions (Dry)
or fully saturated with freshwater (0 ppt) or saltwater (2.5 and 5.0 ppt) and with (+ Wood) or without addition of $^{13}$C-depleted wood.
Standard errors of the mean are in parenthesis (n=4). Data from wood-free and wood-amended soils were analyzed separately. Values
followed by different superscript lowercase letters are significantly different between the four treatments of the wood-free or wood-
amended soils ($P < 0.05$).

| Treatment | Post-SOC Concentration (g kg$^{-1}$) | Post-SOC $\delta^{13}$C (‰) | Wood-derived SOC (%) |
|---|---|---|---|
| Dry | 495 (1.5)[b] | -29.5 (0.20)[a] | . |
| 0 ppt | 493 (3.3)[b] | -29.5 (0.18)[a] | . |
| 2.5 ppt | 488 (4.9)[b] | -29.5 (0.20)[a] | . |
| 5.0 ppt | 460 (8.6)[a] | -29.5 (0.16)[a] | . |
| | | | |
| Dry + Wood | 491 (4.7)[ab] | -30.4 (0.30)[a] | 8 (2.5) |
| 0 ppt + Wood | 502 (4.6)[a] | -30.7 (0.22)[a] | 12 (0.4) |
| 2.5 ppt + Wood | 477 (4.9)[bc] | -30.6 (0.35)[a] | 10 (1.4) |
| 5.0 ppt + Wood | 470 (4.6)[c] | -30.4 (0.14)[a] | 10 (2.0) |


Table 3. Results (F-values and significance) from the repeated measures ANOVA of pH, Eh, microbial biomass C (MBC), $\delta^{13}C$ isotopic signature of MBC, $\delta^{13}CO_2$, and $\delta^{13}CH_4$ measured in soils collected from a coastal freshwater forested wetland and incubated in the laboratory for 98 d under fully saturated conditions with either freshwater or saltwater (2.5 ppt and 5.0 ppt). Data from wood-free and wood-amended soils were analyzed separately.

| Source | pH | Eh | MBC | MBC $^{13}C$ | $\delta^{13}CO_2$ | $\delta^{13}CH_4$ |
|---|---|---|---|---|---|---|
| **Wood-Free** | | | | | | |
| Treatment | 26.6*** | 4.5* | 3.7* | 3.2* | 351.7*** | 60.5*** |
| Time | 4.4*** | 40.7*** | 40.9*** | 15.8** | 24.2*** | 8.3*** |
| Treatment x Treatment | 1.22 | 3.7*** | 27.3*** | 3.3* | 6.4*** | 1.1 |
| | | | | | | |
| **Wood-Amended** | | | | | | |
| Treatment | 29.0*** | 13.6*** | 39.9*** | 2.6 | 129.8*** | 0.3 |
| Time | 18.3*** | 30.1*** | 111.0*** | 3.7 | 34.8*** | 1.4 |
| Treatment x Treatment | 1.4 | 3.4*** | 24.2*** | 5.5** | 8.3*** | 1.0 |

$^*P < 0.05, ^{**}P < 0.01, ^{***}P < 0.0001$

Table 4. Results (F-values and significance) from the one-way ANOVA of cumulative gas production and extracellular enzyme activity (BG: β-glucosidase; PER: peroxidase; NAGase: glucosaminidase; AP: alkaline phosphatase; and AS: arylsulfatase) from soils collected from a coastal freshwater forested wetland and incubated in the laboratory for 98 d under dry conditions or fully saturated conditions with either freshwater or saltwater (2.5 ppt and 5.0 ppt). Data from wood-free and wood-amended soils were analyzed separately.

| Source | $CO_2$ | $CH_4$ | BG | PER | NAGase | AP | AS |
|---|---|---|---|---|---|---|---|
| **Wood-Free** | | | | | | | |
| Treatment | 20.4*** | 15.6*** | 7.2** | 11.9** | 9.5** | 0.9 | 15.8** |
| | | | | | | | |
| **Wood-Amended** | | | | | | | |
| Treatment | 13.3** | 36.7*** | 16.6** | 2.5 | 32.0*** | 2.3 | 31.2*** |

*$P < 0.05$, **$P < 0.01$, ***$P < 0.0001$

Table 5. Initial (1 d) and final (98 d) microbial biomass C (MBC) (mg kg$^{-1}$), MBC $\delta^{13}$C (‰), wood-derived MBC (%) (estimated using $^{13}$C two pool mixing model), and cumulative extracellular enzyme activity (µmol g$^{-1}$) (BG: β-glucosidase; PER: peroxidase; NAGase: glucosaminidase; AP: alkaline phosphatase; and AS: arylsulfatase) for soils incubated under dry conditions (Dry) or saturated conditions with freshwater (0 ppt) or saltwater (2.5 and 5.0 ppt) and with (+ Wood) or without addition of $^{13}$C-depleted wood. Standard errors of the mean are in parenthesis (n=4). Values followed by different superscript lowercase letters are significantly different between the four treatments for the wood-free or wood-amended soils ($P < 0.05$).

| Treatment | Initial MBC Concentration (mg kg$^{-1}$) | Final MBC Concentration (mg kg$^{-1}$) | Initial MBC $\delta^{13}$C (‰) | Final MBC $\delta^{13}$C (‰) | Wood-derived MBC (%) | BG | PER | NAGase | AP | AS |
|---|---|---|---|---|---|---|---|---|---|---|
| Dry | 2238 (400)[c] | 4077 (387)[a] | -27.0 (0.43)[a] | -28.4 (0.28)[ab] | . | 547 (37)[a] | 176 (14)[a] | 240 (20)[a] | 7599 (1038)[a] | 47 (2)[a] |
| 0 ppt | 3982 (196)[ab] | 2657 (344)[b] | -27.3 (0.19)[a] | -28.9 (0.16)[a] | . | 479 (18)[ab] | 197 (38)[a] | 194 (11)[ab] | 6308 (517)[a] | 47 (8)[a] |
| 2.5 ppt | 7334 (1177)[a] | 2495 (195)[b] | -27.8 (0.51)[a] | -27.9 (0.03)[ab] | . | 389 (33)[b] | 412 (75)[b] | 159 (9)[b] | 6539 (183)[a] | 19 (3)[b] |
| 5.0 ppt | 6483 (104)[ab] | 2114 (135)[b] | -27.0 (0.30)[a] | -27.4 (0.15)[b] | . | 379 (27)[b] | 490 (30)[b] | 154 (8)[b] | 6387 (529)[a] | 15 (2)[b] |
| | | | | | | | | | | |
| Dry + Wood | 4444 (579)[a] | 5174 (249)[a] | -29.3 (0.40)[a] | -32.1 (0.44)[a] | 31 (4.9)[a] | 554 (37)[a] | 243 (22)[a] | 275 (17)[a] | 7247 (887)[a] | 40 (2)[a] |
| 0 ppt + Wood | 5376 (330)[a] | 1832 (102)[b] | -29.8 (0.37)[a] | -29.4 (0.15[b] | 4 (1.1)[b] | 349 (24)[b] | 275 (44)[a] | 153 (11)[b] | 4965 (459)[a] | 36 (3)[a] |
| 2.5 ppt + Wood | 5173 (405)[a] | 748 (124)[c] | -30.1 (0.25)[a] | -30.4 (0.95)[ab] | 21 (7.8)[a] | 368 (12)[b] | 365 (30)[a] | 150 (6)[b] | 5548 (653)[a] | 14 (3)[b] |
| 5.0 ppt + Wood | 2123 (400)[b] | 790 (87)[c] | -29.9 (0.43)[a] | -29.7 (0.37)[b] | 18 (1.9)[ab] | 369 (13)[b] | 326 (38)[a] | 150 (6)[b] | 5893 (495)[a] | 13 (2)[b] |

Figure 1. Location of the Alligator River National Wildlife Refuge (ARNWR) within North America (left panel, indicated by box) and location of ARNWR within eastern North Carolina, USA and surrounding freshwater (Alligator River and Albermarle Sound) and saltwater (Pamlico Sound, Croatan Sound, and Roanoke Sound) bodies (right panel). Dotted arrows indicate the location of important surrounding water bodies. The star represents the approximate location of soil and freshwater (from Milltail Creek) sampling locations within the freshwater forested wetlands of ARNWR. The black circle represents the approximate location of saltwater sampling (at the Melvin Daniels Bridge, Roanoke Sound) from the Roanoke Sound. The saltwater was sampled approximately 20 miles east of the soil and freshwater samples.

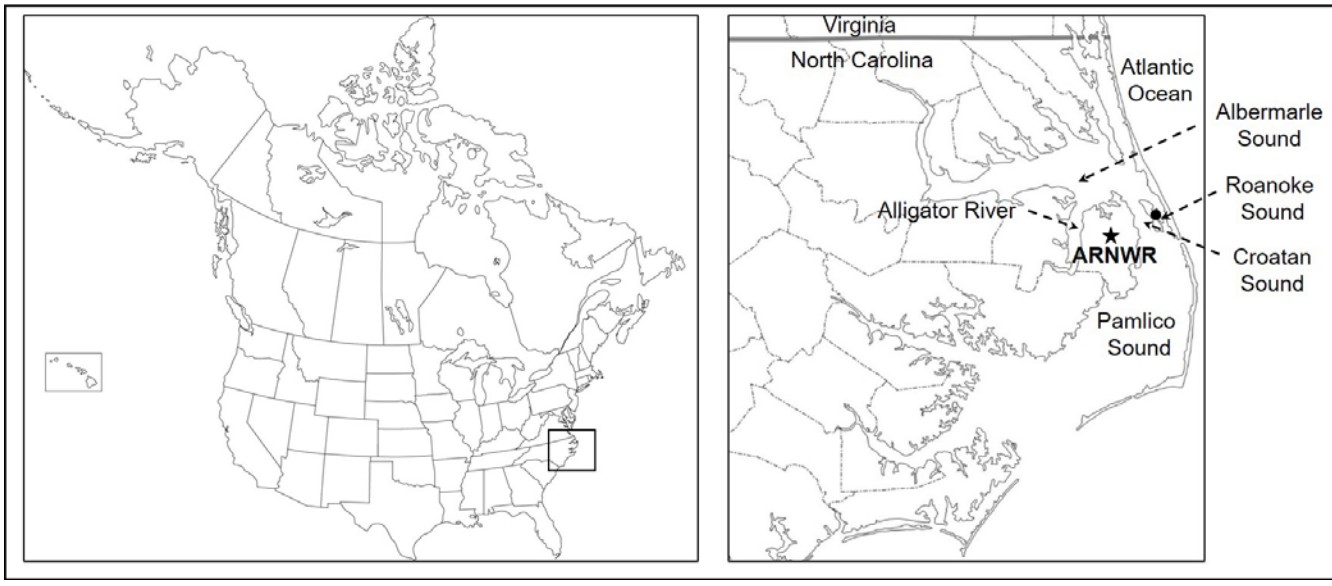

Figure 2. pH for wood-free soils (A) and wood-amended soils (B) and redox potential for wood-free soils (C) and wood-amended soils (D) measured over the course of the 98 d laboratory incubation. Symbols represent mean with standard error (n=4). An asterisk (P < 0.05) indicates significant differences between treatment means at each time point.

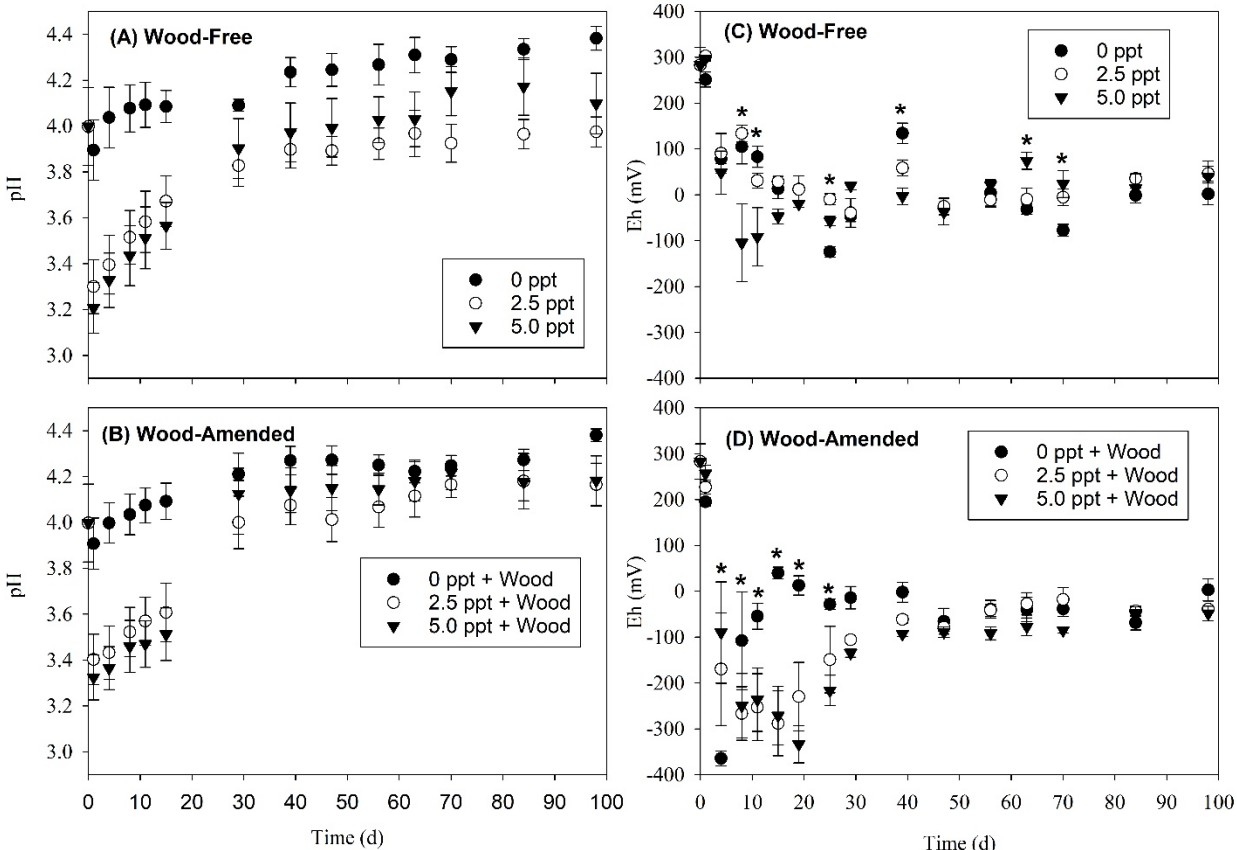

Figure 3. Cumulative $CO_2$ production from wood-free soils (A), wood-amended soils (B), and the wood-associated $CO_2$ production (C); and cumulative $CH_4$ production for wood-free soils (D), wood amended soils (E), and the wood-associated $CH_4$ production (F). Panels C and F refer to the difference between wood-amended and wood-free soils. Bars represent mean with standard error (n=4). Bars with different uppercase letters are significantly different ($P < 0.05$).

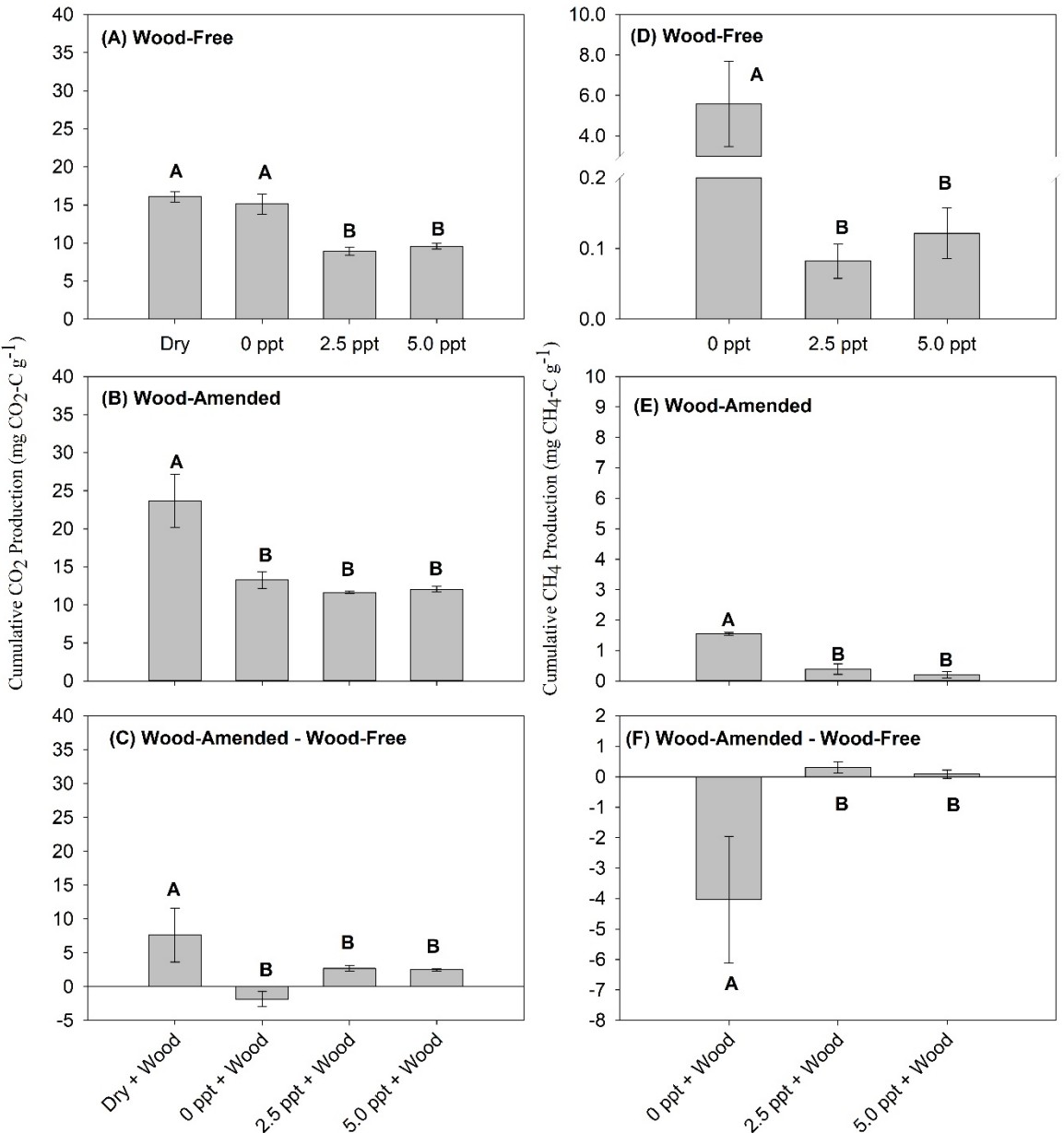

Figure 4. The $\delta^{13}CO_2$ values measured over the course of the 98 d laboratory incubation for wood-free soils (A), wood-amended soils (B), and the proportion of wood-derived $CO_2$ (C). Bars represent mean with standard error (n=4). An asterisk (P < 0.05) indicates significant differences between treatment means at each time point.

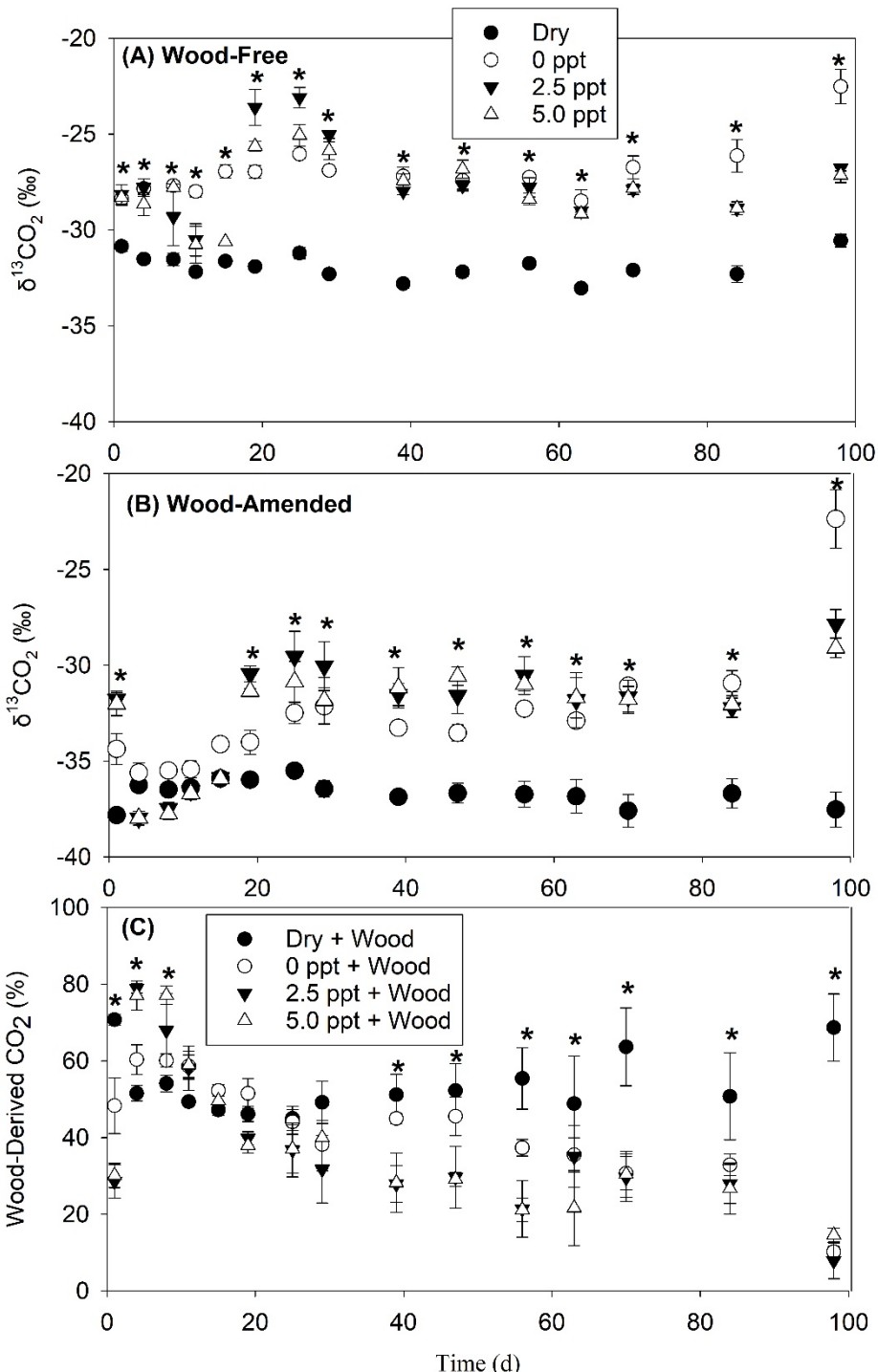

Figure 5. The $\delta^{13}CH_4$ values measured over the course of the 98 d laboratory incubation for wood-free soils (A) and wood-amended soils (B) and the average $\delta^{13}CH_4$ across the entire incubation for wood-free soils (C) and wood-amended soils (D). Symbols or bars represent mean with standard error (n=4). Treatment means with different lowercase letters are significantly different within a sampling time point ($P < 0.05$).

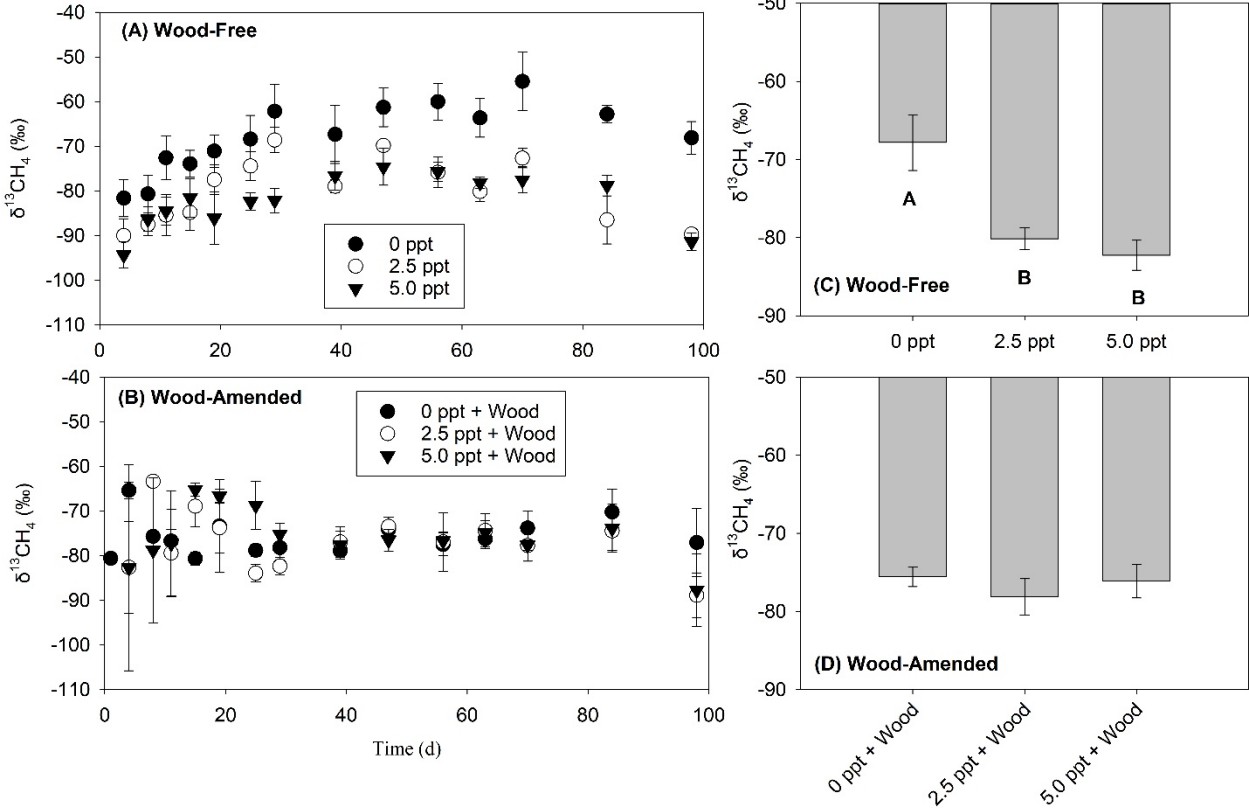

Figure 6. Wood-associated (wood-amended – wood-free) enzyme activity (BG: β-glucosidase; PER: peroxidase; NAGase: glucosaminidase; AP: alkaline phosphatase; and AS: arylsulfatase). Bars represent mean with standard error (n=4). Treatment means with different upper letters are significantly different (*P* < 0.05).

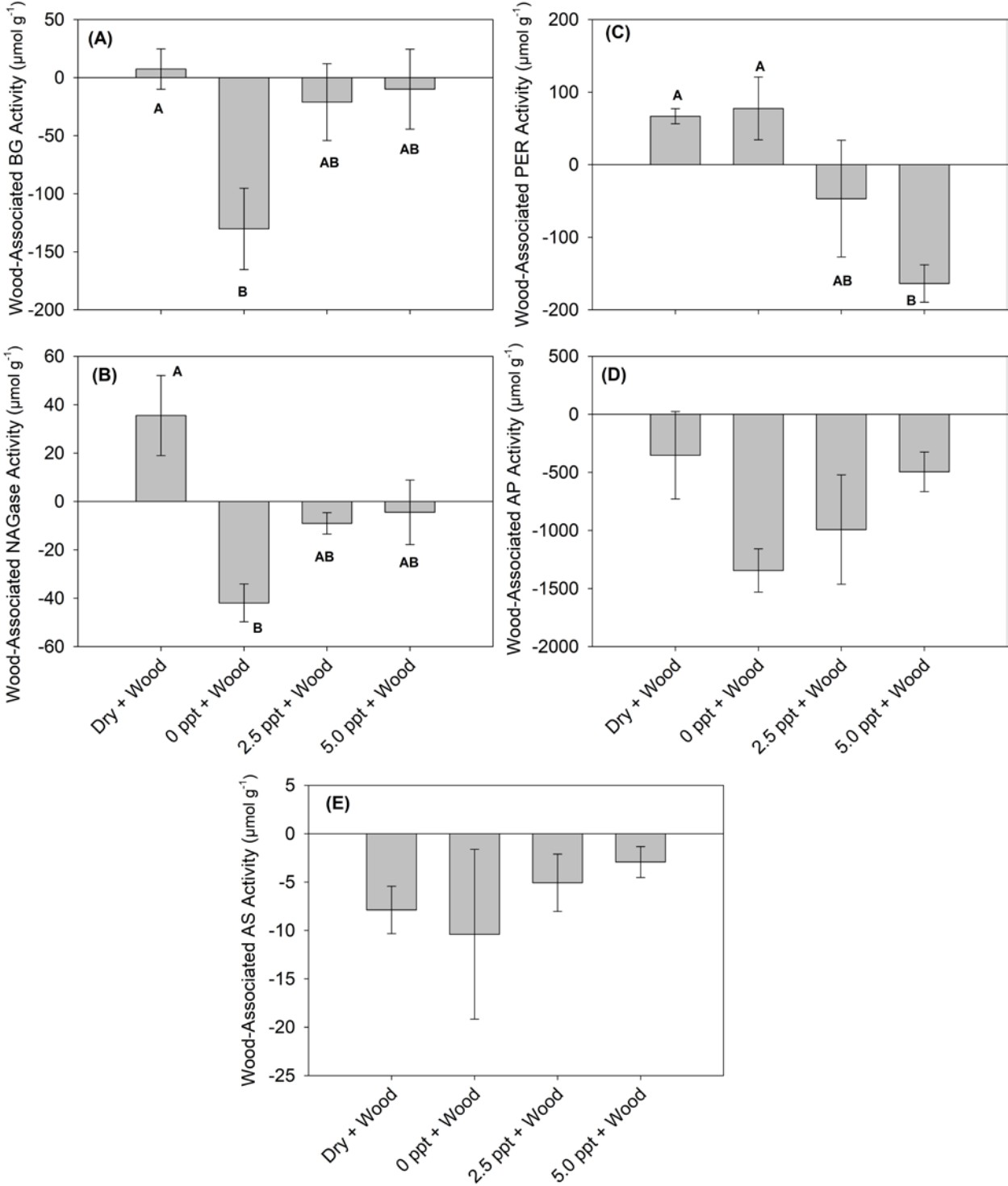