# Peer review of "Saltwater reduces potential $CO_2$ and $CH_4$ production in peat soils from a coastal freshwater forested wetland Kevan J. Minick[a]*, Bhaskar Mitra[b], Asko Noormets[b], John S. King[a]"

_Biogeosciences, 2019_

## Referee Comment (RC1) · Anonymous Referee #1 · 21 Jun 2019

General Comments The authors recognize the threat of saltwater intrusion caused by sea-level rise on non-tidal coastal forests and, using laboratory incubations, test whether additions of salt and coarse woody debris (CWD) change biogeochemical and microbial outputs. They find, among other factors, that salt water reduces total and soil organic carbon and microbial biomass, increases general seawater ions ($SO_4$, Na, Cl, $NH_4$, $NO_3$, $PO_4$, Ca, Mg, K), and over time, and stabilizes pH and Eh more quickly in the presence of CWD. Some enzymatic activity shifts, especially with coarse woody addition, d13C effects are largely unchanged with CWD but significant effects in absence of CWD. Cumulative CO2 and CH4 emissions are reduced with salt, but CWD with FW addition only stimulates CH4 production.

As noted, there is not a large literature on seawater intrusion into these non-tidal systems (I suspect because tidal systems will experience salt intrusion first, thus are the more timely systems of concern), but the postulated scenarios are reasonable, thus providing relative insights into responses of these systems. I appreciate the synthetic discussion and request a few details in my comments to help the reader advance from point to point in the same way the authors have.

1. Does the paper address relevant scientific questions within the scope of BG? yes 2. Does the paper present novel concepts, ideas, tools, or data? I'm not sure about novelty 3. Are substantial conclusions reached? 4. Are the scientific methods and assumptions valid and clearly outlined? I'd like to see hypotheses clearly stated 5. Are the results sufficient to support the interpretations and conclusions? Yes, with some specific clarifications requested 6. Is the description of experiments and calculations sufficiently complete and precise to allow their reproduction by fellow scientists (traceability of results)? yes 7. Do the authors give proper credit to related work and clearly indicate their own new/original contribution? adequate 8. Does the title clearly reflect the contents of the paper? I think so, but a comment included below seems to contradict the title and Figure 2 9. Does the abstract provide a concise and complete summary? yes 10. Is the overall presentation well structured and clear? yes 11. Is the language fluent and precise? Yes, with some subject-verb agreement errors and a few run-on sentences (L83) [There are many cases where subject-verb agreement is not in alignment. e.g. L280 activity. . .were should be activity. . .was; L299 "enzyme . . . were" should be enzyme . . . was] L317 should be "a" one-way ANOVA, no? 12. Are mathematical formulae, symbols, abbreviations, and units correctly defined and used? yes 13. Should any parts of the paper (text, formulae, figures, tables) be clarified, reduced, combined, or eliminated? Might include some of the data driving equations in supplemental sections 14. Are the number and quality of references appropriate? perhaps 15. Is the amount and quality of supplementary material appropriate? No supplemental received

Specific comments Introduction I would have preferred to see clear hypotheses outlined in the last paragraph of the Introduction. The next to last paragraph reads more like Methods to me

Methods I cannot speak in depth to the methods used for isotopic analyses or microbial enzymatic processes. The authors do not disclose the methods used by the NCSU laboratory for the samples they sent to that unit for analysis; I would prefer they do (presumably ion chromatography, and NDIR?). Have the authors any general physico-chemical descriptions of the field soils from where the incubation matrix was collected to help contextualize the work? It seems that other terminal electron acceptors (specifically nitrate) would be useful covariates across the plots that might affect whether a system reaches sulfate reduction, perhaps. It isn't essential that this be provided, but I suggest an interesting consideration if the data are available... The temperature and precipitation data provided are useful (L152), but I'd also like to see the range of these values since over such a long timespan. Might the authors comment on the saltwater treatment levels they selected? These are rather high for a non-tidal system, and the high treatment would be oligohaline in a tidal system. Have levels this high been seen in some nearby areas? L210: Please allay any concerns of positive pressure effects in the chambers during the $\sim$2week intervals between sampling toward the end of the incubation.

Results L339-341: The authors fall into a common trap suggesting that even though a mean is of a different magnitude, that the results vary. They do not. The statistics do not support that wood-amended soils were depleted – the statistics suggest equivalency if all of them are denoted with an "a". (and discussion) Figure 2. I'd like to see something in the discussion related to the pattern of CO2:CH4 reported in the Results. The trend in wood free is parabolic but linear upward in wood-amended. Is that useful? Does this suggest that there an optimal ratio of CWD and salinity that might be targeted to minimize GHG emissions as sea-level rises? L396+: I believe this interpretation follows the same trap noted in L339-341. It is accurate to say MBC was lowest in the dry

treatment across un-amended treatments and lowest in the 5ppt amended treatments.

Discussion L424-425: what C cycling processes are the authors suggesting balance out the reductions in CO2 & CH4? L426 & Figure 2L: I must be missing something, so I suspect other readers will as well. Panels B & E show that the wood-amended plots drop CO2 and CH4 with salt water addition (+2.5 & +5.0 ppt), but the text says it enhances CH4 under saltwater additions. Can you provide clarity? If this is actually referring to the difference (panels C & mostly F), then it seems that the CH4 emissions with CWD are essentially on balance (at the 0 line), no? I've interpreted that saltwater is different than freshwater amendment (A vs B), but the saltwater additions seem to cross the 0 line with the variance. L432: the sentence is almost verbatim earlier in the manuscript (L154). Please revise so each occurrence is unique and not redundant Minor quibble: the hydroperiod operates constantly. I suggest these systems RESPOND over short time scales, but to state they operate on short time scales seems a bit misleading. Even no water is reflected in the hydroperiod in some way, isn't it?

Technical corrections (in addition to a few pointed out previously) L126: The sentence beginning on L126 ("Although many studies...") is unnecessary. That statement was clearly outlined previously in the introduction and does not narrow their research into what they will test and what they expect to find (via the recommended hypotheses addition).L142: why note 13 plots if you only used 4? L199: what year were the trees harvested? L202 & L204: are the 6 rings mentioned in 204 the mean of the 5-7 rings in 202? L248: add (MBC) after spelling out microbial biomass C L286: enzyme XYL is not defined in the 5 above L385: please be more precise than "the last couple" L421: recommend authors use the defined abbreviation "SLR" instead of sea-level rise (else, why define it earlier?) L466: over time (add space) Table 1: please provide units of the ions Figure 2: Please confirm that the labels for panels B & E follow those of C & F (and not A & D). Would you consider a different title for panels C & F? It took me a while to understand that you were reporting the DIFFERENCE between the two, and it wasn't some sort of range (the hyphen notation threw me off). Perhaps "Difference

between wood-amended and wood-free"?

---

## Referee Comment (RC2) · Friederike Gründger (Referee) · 2 Jul 2019

My comments refer to the version of the manuscript that was uploaded by Kevan Minick at 10 May 2019.

The authors present a study that shows the influences of saltwater on CO2 production and CH4 formation processes in non-tidal freshwater-forested wetlands. Soil samples were collected from seven sites located in the Alligator River National Wildlife Refuge (ARNWR) in Dare County, North Carolina, a non-tidal pocosin wetland area that will be most likely effected by sea level rise and saltwater intrusion in the future. The study is based on laboratory incubation experiments testing the effects of freshwater, saltwater and added wood on soil microbial processes in freshwater forested wetland soils. Basic geochemistry, CO2 and CH4 concentrations in incubations, isotopic signatures, microbial biomass carbon measurements, and extracellular enzyme analysis were carried out. The authors confirm that saltwater intrusion can result in reductions in CO2 and CH4 fluxes. Further, they found that coarse woody debris input to soils might reduce CH4 emissions under freshwater conditions, but enhance CO2 production and CH4 emissions under saltwater conditions. The authors also discuss shifts between hydrogenotrophic and acetoclastic methanogenesis dependent on certain incubation conditions.

Please note, that I can't comment on the validity and applicability of the methods used for the analysis of microbial biomass carbon and extracellular enzymes, because I am not an expert in that field.

**General comments:**

1. I wonder why and how soil samples were stored for such a long time (7 weeks) before initiating the incubation experiments. What were the conditions of storage – light, moisture level, oxygen availability? I'd assume that surface soil from hummocks is oxygenated, isn't it? Were the samples kept oxygenated during storage and, if yes, how? What was the temperature at the time of sampling? Only the mean annual temp. is given here. What were the incubation conditions –e.g. temperature, oxygen, volume of incubation? The incubation setup should definitely be more detailed.

2. How far/close were the sampling sites from each other? Would it be useful to add a map that shows soil, freshwater and saltwater sampling sites or pictures of the sampling site and the sampling procedure? I can't imagine the procedure of removing seven 10x10 cm-2 monoliths from hummocks to the depth of the root mat.

3. Fresh- and saltwater were mixed together to get the desired salt concentration for the saltwater treatments. That means, if the water samples weren't sterile-filtered, microbial communities from two different habitats were introduced to the soil microbiota

in the incubations. The same applies to the addition of non-sterilized wood. In the manuscript, microbial interactions due to mixing of samples aren't discussed. As I understand it, the incubations were held under oxic conditions (L213 "flushed at 20 psi for three minutes with CO2/CH4 free zero air). Would it be informative to explain how an aerobic incubation turns into an anoxic environment that promotes methanogenic processes? Also, the sequence of microbial processes that happen along the incubation time and the involvement of certain microbial groups in CO2/CH4 production could be emphasized more detailed.

4. Does the storage of the soil samples under 4°C for 7 weeks caused a shift in microbial community composition and activity already, assuming that in situ temperature at the time of sampling were higher (quick online check for Feb 6 2018, Raleigh, North Carolina, shows 14°C at noon)?

5. Why were these five extracellular enzymes picked to be analyzed? A short description of what these enzymes are catalyzing and in what processes (with regards to your incubations) they are involved would help to understand the concept of the data acquisition (like in L299). Please, add measuring techniques for NAGase, AP, and AS!

6. Can you add a few thoughts about what it means to the environment and climate when CO2/CH4 production increases/decreases due to sea level rise in such areas? e.g. "Findings from this study indicate that substantial changes in the greenhouse gas flux" - how does it change - increase/decrease? What happens to the environment when dead trees provide a significant source of C to already C-rich peat soils? What do we have to aspect after such a change? And why is it important to know what type of methanogenesis is dominant after saltwater intrusion? I am missing the wider picture of the impact of these processes e.g. (L439-442) what are the "important implications for above- and below-ground C cycling dynamics" in particular.

7. I find it a bit difficult to follow the discussion. You start nicely with an overview of your outcomes and the message is clear here. Then you discuss 'CO2 production'

СЗ

results, followed by 'CH4 production' results (L445-456) and the 'competition of the two methanogenic pathways' (L457-476). I suggest, at that point, continuing with the 'isotope section' where different methanogenic pathways are discussed (from L505 on) and then bridge to the 'addition of wood part' (from L480 on). Further, it would help a lot to add a conceptional illustration as final figure showing the possible environmental changes at non-tidal freshwater-forested wetlands after a sea level rise scenario based on your results.

Detailed comments:

L143 Why are only 4 plots used for that study? Isn't it redundant to mention that 13 plots were sampled, if only 4 were used for the study?

L184 instead of: 4) soils incubated at 100% WHC with 5.0 ppt (5.0 ppt). correct to: 4) soils incubated at 100% WHC with saltwater (5.0 ppt).

L199 "dried at to a constant moisture level" – what does that mean? All cookies finally had similar moisture levels or were they dried until moisture per cookie didn't change any longer?

L200 Are "control (non-fertilized) trees" different from the harvested trees that are mentioned before? Is it important to mention that they are non-fertilized? If this information isn't crucial, remove that sentence.

L221 How much soil exactly was removed from the incubation?

L233 With "initial soil samples" you mean the soil that was stored at 4°C for 4 week before the incubation experiment started or homogenized soil samples directly after sampling? Better define the term at some point in 2.3 incubation setup.

L240 "Soil pH was measured on fresh soil samples" – what is meant by fresh soil sample? Soil directly after sampling or after 7 weeks of storage? Instead of using the phrase "fresh", better find a term that clearly describes the condition of the sample (same for L250).

L250 Avoid the term "fresh soil" when it was a soil subsample from an incubation. Fresh soil is anyway not a precise definition of a condition of a soil.

L279 change into: enzymes were quantified on soil samples on days 0, 1, 8, 35, and 98 of the soil incubation.

L285 Can't find enzyme XYL in the description of measured enzymes above.

L383 "while the proportion of wood-derived CO2 remained steady for a good portion of the incubation but increased in the final couple measurements periods" – add something that indicates that you are referring to dry incubations. "for a good portion" and "final couple" isn't precise enough. Add proper terms for time scales.

L433 When parameters like the redox potential in an incubation were measured, then it isn't called "in situ" measurement. In situ would be, when the measurements were done at the ARNWR sampling site. If the values shown are indeed in situ measurements, why aren't they mentioned in the result part? At least, I can't find them there.

L458 "Numerous others studies have found that saltwater reduces CH4 fluxes compared to freswater, both within the field and laboratory." – add references. Correct typo in freshwater.

---

## Author Comment (AC1) · 18 Jul 2019

Sorry for the delayed response! The authors and co authors responses are given below and appear following the reviewers comments. We will upload the revised manuscript once we have completed responses to the second reviewers comments.
General Comments The authors recognize the threat of saltwater intrusion caused by sea-level rise on non-tidal coastal forests and, using laboratory incubations, test whether additions of salt and coarse woody debris (CWD) change biogeochemical and microbial outputs. They find, among other factors, that salt water reduces total and soil organic carbon and microbial biomass, increases general seawater ions (SO4, Na, Cl, NH4, NO3, PO4, Ca, Mg, K), and over time, and stabilizes pH and Eh more quickly in the presence of CWD. Some enzymatic activity shifts, especially with coarse woody addition, d13C effects are largely unchanged with CWD but significant effects in absence of CWD. Cumulative CO2 and CH4 emissions are reduced with salt, but CWD with FW addition only stimulates CH4 production.

As noted, there is not a large literature on seawater intrusion into these non-tidal systems (I suspect because tidal systems will experience salt intrusion first, thus are the more timely systems of concern), but the postulated scenarios are reasonable, thus providing relative insights into responses of these systems. I appreciate the synthetic discussion and request a few details in my comments to help the reader advance from point to point in the same way the authors have.

1. Does the paper address relevant scientific questions within the scope of BG? yes 2. Does the paper present novel concepts, ideas, tools, or data? I'm not sure about novelty 3. Are substantial conclusions reached? 4. Are the scientific methods and assumptions valid and clearly outlined? I'd like to see hypotheses clearly stated 5. Are the results sufficient to support the interpretations and conclusions? Yes, with some specific clarifications requested 6. Is the description of experiments and calculations sufficiently complete and precise to allow their reproduction by fellow scientists (traceability of results)? yes 7. Do the authors give proper credit to related work and clearly indicate their own new/original contribution? adequate 8. Does the title clearly reflect the contents of the paper? I think so, but a comment included below seems to contradict the title and Figure 2 9. Does the abstract provide a concise and complete summary? yes 10. Is the overall presentation well structured and clear? yes
11. Is the language fluent and precise? Yes, with some subject-verb agreement errors and a few run-on sentences (L83) [There are many cases where subject-verb agreement is not in alignment. e.g.

L280 activity. . .were should be activity. . .was;

This has been corrected

L299 "enzyme . . . were" should be enzyme . . . was]

This has been corrected

L317 should be "a" one-way ANOVA, no?

An "a" has been added

12. Are mathematical formulae, symbols, abbreviations, and units correctly defined and used? yes 13. Should any parts of the paper (text, formulae, figures, tables) be clarified, reduced, combined, or eliminated? Might include some of the data driving equations in supplemental sections

These are very common equations and are not necessary to include.

14. Are the number and quality of references appropriate? perhaps 15. Is the amount and quality of supplementary material appropriate? No supplemental received Introduction:

I would have preferred to see clear hypotheses outlined in the last paragraph of the Introduction. The next to last paragraph reads more like Methods to me

We have added objectives to this section and changed the wording to sound less like methods.

Methods:

I cannot speak in depth to the methods used for isotopic analyses or microbial en-
zymatic processes. The authors do not disclose the methods used by the NCSU laboratory for the samples they sent to that unit for analysis; I would prefer they do (presumably ion chromatography, and NDIR?).

We have added this information.

Have the authors any general physicochemical descriptions of the field soils from where the incubation matrix was collected to help contextualize the work? It seems that other terminal electron acceptors (specifically nitrate) would be useful covariates across the plots that might affect whether a system reaches sulfate reduction, perhaps.

We measured soil ions in a study from 2013, using ion exchange probes (PRS probes) (Minick et al. 2019). These probes collected anions and cations over one six week period from July to August 2013 in the same plots used for this study. NO3- concentrations were very low and likely contributed little contributed little to the potential pool of electron acceptors. Alternatively S and Fe availability were much higher than NO3- in the hummocks, as measured using the same PRS probes. Given that the soils were completely saturated (e.g. flooded) with either fresh or salt water, and numerous ions were measured (regrettably not Fe though), we feel this represents an acceptable

Minick, K. J., Kelley, A. M., Miao, G., Li, X., Noormets, A., Mitra, B., and King, J. S.: Microtopography alters hydrology, phenol oxidase activity and nutrient availability in organic soils of a coastal freshwater forested wetland, Wetlands 39, 263-273, https://doi.org/10.1007/s13157-018-1107-5, 2019a.

It isn't essential that this be provided, but I suggest an interesting consideration if the data are available... The temperature and precipitation data provided are useful (L152), but I'd also like to see the range of these values since over such a long timespan.

We have kept this section as is.

Might the authors comment on the saltwater treatment levels they selected? These are rather high for a non-tidal system, and the high treatment would be oligonaline in a tidal
system. Have levels this high been seen in some nearby areas?

Yes, the saltwater in the sound to the east (only a mile or so) ranges from approximately 1-5 percent saltwater, another couple miles into the sound and towards the ocean the water is up to 10-20 percent. So these values are reasonable. The 2.5 percent is a more likely, or relatively short term scenario, while the 5 percent represents a more extreme or long term scenario.

L210: Please allay any concerns of positive pressure effects in the chambers during the âĹij2week intervals between sampling toward the end of the incubation.

The lids were left loose between sampling periods. This is stated in the following lines (L211-212).

Results:

L339-341: The authors fall into a common trap suggesting that even though a mean is of a different magnitude, that the results vary. They do not. The statistics do not support that wood-amended soils were depleted – the statistics suggest equivalency if all of them are denoted with an "a". (and discussion)

The end of the sentence has been removed.

Figure 2. I'd like to see something in the discussion related to the pattern of CO2:CH4 reported in the Results. The trend in wood free is parabolic but linear upward in wood-amended. Is that useful? Does this suggest that there an optimal ratio of CWD and salinity that might be targeted to minimize GHG emissions as sea-level rises?

Our experiment was not intended to determine different levels of CWD inputs with all incubations receiving the proportionally same amount of wood additions and so we cannot test the combination of varying effects of wood and salinity. With that said, we think the reviewers observation is a good one and worth noting. We have added discussion on the CO2:CH4 trend in the discussion to this specifically

BGD
L396+: I believe this interpretation follows the same trap noted in L339-341. It is accurate to say MBC was lowest in the dry treatment across un-amended treatments and lowest in the 5ppt amended treatments.

We have made these changes

Discussion:

L424-425: what C cycling processes are the authors suggesting balance out the reductions in CO2 & CH4?

We have clarified this sentence

L426 & Figure 2L: I must be missing something, so I suspect other readers will as well. Panels B & E show that the wood-amended plots drop CO2 and CH4 with salt water addition (+2.5 & +5.0 ppt), but the text says it enhances CH4 under saltwater additions. Can you provide clarity? If this is actually referring to the difference (panels C & mostly F), then it seems that the CH4 emissions with CWD are essentially on balance (at the 0 line), no? I've interpreted that saltwater is different than freshwater amendment (A vs B), but the saltwater additions seem to cross the 0 line with the variance.

We have added clarification in the text to address this potential confusion. Panels C and F show the difference between wood free and wood amended soils which gives the wood-associated CO2 and CH4 production. So within the wood free or wood amended treatments, salt water generally reduced gas production. But when comparing wood free and wood amended gas fluxes for each specific gas, we actually see that wood additions highly reduced CH4 from freshwater but enhanced it in salt water incubations. This is just another way of looking at the results in order to derive some interpretation of how wood versus non wood treatments influence gas production when incubated with fresh or salt water.

L432: the sentence is almost verbatim earlier in the manuscript (L154). Please revise so each occurrence is unique and not redundant Minor quibble: the hydroperiod

BGD
operates constantly. I suggest these systems RESPOND over short time scales, but to state they operate on short time scales seems a bit misleading. Even no water is reflected in the hydroperiod in some way, isn't it? Technical corrections (in addition to a few pointed out previously)

We have made changes to the sentence in the discussion and changed operates to responds. We agree with the reviewers assessment

L126: The sentence beginning on L126 ("Although many studies. . .") is unnecessary. That statement was clearly outlined previously in the introduction and does not narrow their research into what they will test and what they expect to find (via the recommended hypotheses addition).

We have made some changes to this paragraph but have kept this sentence because we think it helps guide the reader in this summary introduction paragraph.

L142: why note 13 plots if you only used 4?

We have mentioned the thirteen plots because it is part of the description of the site. We feel it is important to note that this site is part of the Ameriflux network, which follows certain experimental design protocols. Of the thirteen plots, four of these are more intensively monitored for plant and soil processes. We have added information to this sentence to highlight why we chose four plots, to hopefully clarify why we chose to mention this.

L199: what year were the trees harvested?

2010, we added that it was harvested then.

L202 & L204: are the 6 rings mentioned in 204 the mean of the 5-7 rings in 202?

We have revised this section. It was six tree rings. We reduced mentioning it to only once.

L248: add (MBC) after spelling out microbial biomass C

BGD
This correction has been made

L286: enzyme XYL is not defined in the 5 above

This information has been added

L385: please be more precise than "the last couple"

We just removed that part of the sentence, due to it being somewhat subjective and not adding much to the overall results or interpretation

L421: recommend authors use the defined abbreviation "SLR" instead of sea-level rise (else, why define it earlier?)

This has been changed

L466: over time (add space)

This has been changed

Table 1: please provide units of the ions

This information has been added

Figure 2: Please confirm that the labels for panels B & E follow those of C & F (and not A & D). Would you consider a different title for panels C & F? It took me a while to understand that you were reporting the DIFFERENCE between the two, and it wasn't some sort of range (the hyphen notation threw me off). Perhaps "Difference between wood-amended and wood-free"?

We have added a sentence to the figure caption to show this.

---

## Author Comment (AC2) · 19 Jul 2019

We appreciate the reviewers detailed feedback and believe they have improved the manuscript greatly. We have provided responses to the reviewers comments. We hope our revisions are in line with what the reviewer wanted.

Referee #2

Friederike Gründger (Referee) friederike.gruendger@bios.au.dk

My comments refer to the version of the manuscript that was uploaded by Kevan Minick

[Figure]

The authors present a study that shows the inïnĆuences of saltwater on $CO_2$ production and $CH_4$ formation processes in non-tidal freshwater-forested wetlands. Soil samples were collected from seven sites located in the Alligator River National Wildlife Refuge (ARNWR) in Dare County, North Carolina, a non-tidal pocosin wetland area that will be most likely effected by sea level rise and saltwater intrusion in the future. The study is based on laboratory incubation experiments testing the effects of freshwater, saltwater and added wood on soil microbial processes in freshwater forested wetland soils. Basic geochemistry, $CO_2$ and $CH_4$ concentrations in incubations, isotopic signatures, microbial biomass carbon measurements, and extracellular enzyme analysis were carried out. The authors conïnĄrm that saltwater intrusion can result in reductions in $CO_2$ and $CH_4$ ïnĆuxes. Further, they found that coarse woody debris input to soils might reduce $CH_4$ emissions under freshwater conditions, but enhance $CO_2$ production and $CH_4$ emissions under saltwater conditions. The authors also discuss shifts between hydrogenotrophic and acetoclastic methanogenesis dependent on certain incubation conditions. Please note, that I cannot comment on the validity and applicability of the methods used for the analysis of microbial biomass carbon and extracellular enzymes, because I am not an expert in that ïnĄeld.

General comments:

1. I wonder why and how soil samples were stored for such a long time (7 weeks) before initiating the incubation experiments. What were the conditions of storage – light, moisture level, oxygen availability? I'd assume that surface soil from hummocks is oxygenated, isn't it? Were the samples kept oxygenated during storage and, if yes, how? What was the temperature at the time of sampling? Only the mean annual temp. is given here. What were the incubation conditions –e.g. temperature, oxygen, volume of incubation? The incubation setup should deïnĄnitely be more detailed.

Samples were stored as other parts of the experiment were being initiated. Samples

were stored at 4C, in a fridge, in the dark. The samples were stored based on their initial soil moisture levels, which were approximately 90% moisture. The hummocks are somewhat oxygenated but that depends on the water table depth. The hummocks are frequently inundated throughout the year when precip is high.

We have added more detail on the incubation setup in the section 2.3.

2. How far/close were the sampling sites from each other? Would it be useful to add a map that shows soil, freshwater and saltwater sampling sites or pictures of the sampling site and the sampling procedure? I can't imagine the procedure of removing seven 10x10 cm-2 monoliths from hummocks to the depth of the root mat.

We have added a new figure 1, a map with soil and water sampling locations and surrounding water bodies. The soils were sampled within a quarter mile of freshwater. The saltwater was sampled approximately 20 miles east of soil and water samples.

Soils were removed using a saw and cutting in a 10 x 10 cm-2, using a pvc square as guidance. This is in the methods.

2. Fresh- and saltwater were mixed together to get the desired salt concentration for the saltwater treatments. That means, if the water samples weren't sterile-fiAltered, microbial communities from two different habitats were introduced to the soil microbiota in the incubations. The same applies to the addition of non-sterilized wood. In the manuscript, microbial interactions due to mixing of samples aren't discussed.

Samples were filtered through Whatman #2 filters (8 $\mu$m) to remove particulates. This information has been added. This would not sterilize the water from microbial populations by any mean. This mixing of microbial populations from the different water and soil sources were mixed together, although we would argue this represents what would occur during salt water inundation into these freshwater systems, either in short term pulses (such as storm surges) or longer term inundation periods with rising sea levels.

As I understand it, the incubations were held under oxic conditions (L213 "flushed

at 20 psi for three minutes with CO2/CH4 free zero air). Would it be informative to explain how an aerobic incubation turns into an anoxic environment that promotes methanogenic processes? Also, the sequence of microbial processes that happen along the incubation time and the involvement of certain microbial groups in CO2/CH4 production could be emphasized more detailed.

We understand the reviewers concern but argue that our incubations were indeed anaerobic for the following reasons:

1) Although the incubations had oxic headspace (CO2 and CH4 free air, but containing O2), the soils were incubated at 100 % WHC which resulted in soils being completely flooded (either fresh- or salt-water) with water essentially covering the surface of the incubated soils, thereby allowing for the development of anaerobic conditions similar to that observed in the field and for subsequent production of CH4 through the anaerobic process of methanogenesis. We have added that information at the beginning of section 2.3 of the methods. Further, O2 presence in the headspace would diffuse very slowly into the water (rates of O2 diffusion into water is approximately 5,800 to 9,500 times lower than that in water (Massman 1998)) and therefore would likely be of negligible effect on total Ch4 production.

2) We actually took measurements of redox potential throughout the experiment (see Figure 1C and 1D)! This showed that incubations were indeed anaerobic, starting initially at +300 mV and dropping quickly to between approximately 100 and -400 throughout most of the incubation, with the wood additions dropping Eh much lower than non-wood treatments.

3) The rates of CH4 production are quite high, which in and of itself indicate that the incubations were anaerobic. We ran four blank incubations (jars with no soil) that were treated exactly the same (most importantly flushing with same air) and sampled on the same schedule as soil incubations. We have added a couple sentences about the blanks in the methods section. Further, when compared to anaerobic incubations (with

N2 headspace) of soils from northern latitude wetlands, we see that our measurements are much much greater (see Treat et al. 2014; Walz et al. 2017 for instance).

Treat C, Wollheim WM, Varner R, Grandy AS, Talbot J, Frolking S (2014) Temperature and peat type control $CO_2$ and $CH_4$ production in Alaskan permafrost peats. Global Change Biology, 20, 2674-2686.

Walz, J., C. Knoblauch, L. Böhme and E.-M. Pfeiffer (2017). "Regulation of soil organic matter decomposition in permafrost-affected Siberian tundra soils - Impact of oxygen availability, freezing and thawing, temperature, and labile organic matter." Soil Biology and Biochemistry 110: 34-43.

4. Does the storage of the soil samples under 4°C for 7 weeks cause a shift in microbial community composition and activity already, assuming that in situ temperature at the time of sampling were higher (quick online check for Feb 6 2018, Raleigh, North Carolina, shows 14°C at noon)?

It is unlikely that storage temperature and time resulted in a significant shift in microbial communities and/or activity that would affect the results and inference from this experiment, given all samples were treated the same. Storing freshly collected soils at 4°C (refrigerator temperature) is very common in soil microbial studies, and in fact many publications do not even state how long soils were stored before some kind of laboratory procedure! The reasoning is that at such a cold temperature forces the microbes and microbial processes to slow significantly, so that there is minimal decomposition/activity during storage and until incubation. There is also a fair amount of pre-incubation/processing that occurs before incubation of soils in these types of studies, making storage of soils in the most inert way possible (without altering soil biogeochemical conditions as much as possible) necessary in order to complete those tasks before actually starting the incubation. Ideally, it would have occurred around 2-4 weeks post collection but in this case it was not possible.

5. Why were these fiĄve extracellular enzymes picked to be analyzed? A short description of what these enzymes are catalyzing and in what processes (with regards to your incubations) they are involved would help to understand the concept of the data acquisition (like in L299). Please, add measuring techniques for NAGase, AP, and AS!

We have added more information on what substrates/compounds these enzymes degrade. We have added NAGase, AP, and AS to the hydrolytic enzyme assay information.

6. Can you add a few thoughts about what it means to the environment and climate when CO2/CH4 production increases/decreases due to sea level rise in such areas? e.g. "Findings from this study indicate that substantial changes in the greenhouse gas flux" - how does it change - increase/decrease? What happens to the environment when dead trees provide a significant source of C to already C-rich peat soils? What do we have to expect after such a change? And why is it important to know what type of methanogenesis is dominant after saltwater intrusion? I am missing the wider picture of the impact of these processes e.g. (L439-442) what are the "important implications for above- and below-ground C cycling dynamics" in particular.

We have added a few sentences to the conclusion to expand somewhat on implications of this study, but hesitate to speculate too much about how well our lab experiment would represent ecosystem responses on a large scale (a common critique of studies like this in general). We have provided some detail on what to expect (e.g. C inputs to soils, ecosystem transition, etc), as well as suggestions for future directions. What this study does provide is insight into the ecosystem/soil response and provides mechanistic details on why we might find this response. For instance, this is why understanding the pathway of methane formation can be informative. The two different pathways appear to be linked with very different magnitudes of fluxes, with hydrogenotrophic pathway having lower methane production than the acetoclastic pathway.

7. I find it a bit difiňĄcult to follow the discussion. You start nicely with an overview of your outcomes and the message is clear here. Then you discuss 'CO2 production'

results, followed by 'CH4 production' results (L445-456) and the 'competition of the two methanogenic pathways' (L457-476). I suggest, at that point, continuing with the isotope section 'where different methanogenic pathways are discussed (from L505 on) and then bridge to the 'addition of wood part' (from L480 on). Further, it would help a lot to add a conceptual illustration as final figure showing the possible environmental changes at non-tidal freshwater-forested wetlands after a sea level rise scenario based on your results.

We have switched the two paragraphs as suggested.

Detailed comments:

L143 Why are only 4 plots used for that study? Isn't it redundant to mention that 13 plots were sampled, if only 4 were used for the study?

We have mentioned the thirteen plots because it is part of the description of the site. We feel it is important to note that this site is part of the Ameriflux network, which follows certain experimental design protocols. Of the thirteen plots, four of these are more intensively monitored for plant and soil processes. We have added information to this sentence to highlight why we chose four plots, to hopefully clarify why we chose to mention this.

L184 instead of: 4) soils incubated at 100% WHC with 5.0 ppt (5.0 ppt). correct to: 4) soils incubated at 100% WHC with saltwater (5.0 ppt).

"saltwater" was accidentally left out of the description and should have come after "with 5.0 ppt". We have added this information, which also keeps it consistent with treatment "3)" description.

L199 "dried at to a constant moisture level" – what does that mean? All cookies finally had similar moisture levels or were they dried until moisture per cookie didn't change any longer?

The latter, this means that the cookies were dried until no more change in moisture

was measured.

L200 Are "control (non-fertilized) trees" different from the harvested trees that are mentioned before? Is it important to mention that they are non-fertilized? If this information isn't crucial, remove that sentence.

It is not important to mention. These are the same trees that were harvested. Some were from a fertilization treatment and some from a non fertilized treatment. We only used trees from the non fert trt. We have removed that sentence.

L221 How much soil exactly was removed from the incubation?

Approximately 1.0 g dry soil weight was removed at each enzyme sampling date. We have added this information to this sentence.

L233 With "initial soil samples" you mean the soil that was stored at 4◦C for 4 week before the incubation experiment started or homogenized soil samples directly after sampling? Better define the term at some point in 2.3 incubation setup.

The initial samples were removed from the homogenized bag prior to the start of the incubation. We have added this information L240 "Soil pH was measured on fresh soil samples" – what is meant by fresh soil sample? Soil directly after sampling or after 7 weeks of storage? Instead of using the phrase "fresh", better find a term that clearly describes the condition of the sample (same for L250).

This was measured the in soils after storage, the same day the incubations were started. We have added this information here.

L250 Avoid the term "fresh soil" when it was a soil subsample from an incubation. Fresh soil is anyway not a precise definition of a condition of a soil.

Fresh has been removed from this sentence

L279 change into: enzymes were quantified on soil samples on days 0, 1, 8, 35, and 98 of the soil incubation.

We have removed the "(day 0)" from the sentence to better reflect what was done. The measurements at "day 0" were done on soil samples before incubation. Therefore we have removed the "day 0" reference to avoid confusion that these were subjected to the different salt and fresh water treamtents.

L285 Can't find enzyme XYL in the description of measured enzymes above.

This information has been added

L383 "while the proportion of wood-derived CO2 remained steady for a good portion of the incubation but increased in the final couple measurements periods" – add something that indicates that you are referring to dry incubations. "for a good portion" and "final couple" isn't precise enough. Add proper terms for time scales.

We have modified this sentence

L433 When parameters like the redox potential in an incubation were measured, they arent called "in situ" measurement. In situ would be, when the measurements were done at the ARNWR sampling site. If the values shown are indeed in situ measurements, why aren't they mentioned in the result part? At least, I can't find them there.

This is data collected from the field site. It is unpublished data and is not replicated but more observational from testing Eh during frequent field visits as a way to get an idea of what redox potentials we can maybe expect. More detailed studies of in situ redox potential are important and something we are very interested in, but cant provide that at this current time. We can state it is unpublished in parenthesis or leave as is. We feel it is important to mention though.

L458 "Numerous others studies have found that saltwater reduces CH4 fluxes compared to freswater, both within the field and laboratory." – add references. Correct typo in freshwater. This correction has been made

---

## Referee Report (RR1)

Comments on the revised manuscript.

I acknowledge that the authors considered my comments. From my perspective the manuscript did improve very much – methods are better understandable and the discussion is well structured.

Nevertheless, I would appreciate, if the authors would state that the *in situ* measurements of redox potential are unpublished data (add unpublished data in brackets; L467) to avoid confusion between data presented in this study and data that are not shown in the results. Additionally, please, check the manuscript for spelling mistakes and comma placement.
Only a few things that caught my eye:
L2011 (approximately2 g fresh weight) -> (approximately 2 g fresh weight)

L558, 563, 820, 841 salt water -> saltwater

I am not a native speaker, so I only guess that in a term like "Changes in fresh- and salt-water" (e.g. L182, 186) 'salt-water' should be written as 'saltwater' as it was done throughout the manuscript, but I can be wrong. To me, it would make more sense to be consistent. Same with fresh-water in L183.

I state that version 4 of the manuscript with the title "Saltwater reduces potential CO2 and CH4 production in peat soils from a coastal freshwater forested wetland" (MS No.: bg-2019-174) is acceptable for final publication in BG after addressing the comments above.

---

## Author Response (AR2)

*Dear Helge and reviewers,*

*Thank you for considering our manuscript for revision and resubmission following minor revisions. We have carefully considered your and the reviewers' comments and have revised the manuscript accordingly. Our responses to comments are provided in italicized black font below each comment. We also did a careful revision of the manuscript, as well, for consistency in writing style, wording, and correct grammar. The authors have uploaded a revised manuscript changes incorporated, a revised manuscript with changes still visible (as a supplement), as well as this response to reviewer's comments. We appreciate the timely feedback and consideration of this manuscript for publication in Biogeosciences.*

*Thank you,*

*Kevan*

Associate Editor Decision: Publish subject to minor revisions (review by editor) (15 Sep 2019) by Helge Niemann

Comments to the Author:

Dear Kevan Minick,

Your revised version has been checked by the reviewers and although both agree that your MS has improved, there are still some issues open, and I agree here fully with the reviewers.

I am happy to receive a second revision with considerations of the remaining reviewers concerns. Please provide a clear statement with arguments if you chose not to consider a certain concern.

Best wishes, Helge Niemann

**Reviewer 1:**

Comments on the revised manuscript.

I acknowledge that the authors considered my comments. From my perspective the manuscript did improve very much – methods are better understandable and the discussion is well structured.

> *We appreciate the constructive feedback and detailed comments provided in both rounds of revisions. These comments have greatly improved the quality of the manuscript.*

Nevertheless, I would appreciate, if the authors would state that the in situ measurements of redox potential are unpublished data (add unpublished data in brackets; L467) to avoid confusion between data presented in this study and data that are not shown in the results. Additionally, please, check the manuscript for spelling mistakes and comma placement.

> *We have added "unpublished data" to the end of that sentence. We have also checked the manuscript for spelling, comma placement and other grammatical errors pointed out by both reviewers.*

Only a few things that caught my eye:

L2011 (approximately2 g fresh weight) -> (approximately 2 g fresh weight)

*This has been fixed.*

L558, 563, 820, 841 salt water -> saltwater

*We have made these corrections*

I am not a native speaker, so I only guess that in a term like "Changes in fresh- and salt-water" (e.g. L182, 186) 'salt-water' should be written as 'saltwater' as it was done throughout the manuscript, but I can be wrong. To me, it would make more sense to be consistent. Same with fresh-water in L183.
I state that version 4 of the manuscript with the title "Saltwater reduces potential CO2 and CH4 production in peat soils from a coastal freshwater forested wetland"

> *We agree with the reviewer. We have written out freshwater and saltwater in each of the identified instances (as well as a few other places) to be consistent throughout the manuscript.*

**Reviewer 2:**

Thanks to the authors for answering questions and addressing some of the concerns raised by the reviewers. It is regrettable that several concerns were dismissed in their response. My biggest concern at this point is that both reviewers noted that we were not familiar with the methods for isotopic analysis or microbial enzymatic processes, so this – in essence – has not been reviewed at this time. These comments may be shared with the authors.

> *We acknowledge the reviewers concern about dismissal of previous comments. We tried to provide detailed responses and explanations to both reviewer's comments, where appropriate, and did not purposefully dismiss comments by either reviewer. We apologize if we did overlook specific comments.*
>
> *The authors have recently (Minick et al. 2019a, b: in reference list of manuscript) published manuscripts with the same enzyme methods and isotopic analysis. The methods are well established, particularly the enzyme assays. For the isotopic analysis, we have added the equation and reference for the calculation of 13C derived CO2 (Figure 4c), using a two pool isotopic mixing model, as this may not be as well known. We have also added information on solid and gas sample calibrations used for the Picarro instrument and in determining unknown sample C concentration and 13C signature. Hopefully this will help alleviate concerns over this methodology.*

**Comments on response to Reviewer 1**

Reviewer 1, point 13. Apparently, at least one reader wasn't aware of the formula initially, so "not necessary" is arguable

> *We previously provided equations for MBC and 13C calculations of MBC, which are less common. Upon rereading the previous comment, it appears unclear what particular equations the reviewer wanted to see. The other equations to potentially include are those for calculations of enzyme activity and wood derived CO2 using the 13CO2 data. We have 13CO2 calculations as readers may not be as familiar with these. The calculation of enzyme activity was described in writing. This calculation is common and straight forward and therefore is not typically represented as an equation in a manuscript.*

Comments responding to request for physicochemical discussion fall off mid-sentence

> *We apologize for the incomplete answer to this comment. We measured ions in water and have some previous measurements on ion exchangeable elements from a previous study and have also measured soil C pools and 13C signature in different organic and mineral soil horizons at this site. Some of this is referenced in the manuscript, but not directly discussed in relation to this current study. We have added some properties of the soils in the methods that may be of use (C concentration, 13C of SOC, and pH), from Minick et al. 2019b. We also briefly discuss the ion exchangeable data in the introduction (Minick et al. 2019a). Other than those two studies, we do not have any further soil physicochemical data at his date.*

Salinity levels addressed in comments but not the manuscript

> *We have added information about reasoning for the salinity levels chosen to the "sample collection" section of the methods and with reference to Figure 1*

Comments on Figure 2. This reviewer was not suggesting different levels of CWD would or should have been tested. It was an observation that, as the authors note, might be interesting & worthy of further study.

> *In the last revised manuscript, we added a couple of sentences in the discussion pointing out this observation that there may be some interesting interactions between CWD and salt water at different levels.*

**LATEST VERSION**

**Substantive**

L163: I still contend that a mean over such a long period is not very useful without some indicator of variability. I had suggested presenting the range, but std dev or other metrics would be useful.

> *We have added average annual temps and cumulative precip, with standard error for each, from the years 2008 to 2018.*

L241: (no soil) incubations: does that mean it was just water (and maybe CWD) in the jars? This isn't immediately obvious to me (so then likely other readers).

> *These blank incubations had no soil, water, or wood in them. We have added information in the text to clarify what the blank samples represent.*

Why were the masses of samples and dates of measurements changed between versions? This makes me question methodology and data integrity… (L286, L316, L327)

> *When reviewing some of the data, we realized that the initial statement of 0.5 g of dry weight, for enzyme analysis, should be stated as 0.8. The samples range from about 0.7 to 0.9 in dry weight depending on the total amount removed for each measurement period. For MBC, the value should have remained at 0.5, which was initially stated. We have changed this back. We are sorry for the confusion. All of the correct data was used in the calculations of enzyme activity and microbial biomass.*

**Minor**

Authors use Oxford comma in some instances and not others. It should be used or not used consistently throughout the manuscript.

*We have tried to be consistent with our use of commas, including making the suggested changes in the comments below*

The paragraph beginning L89: I find the switch between "sulfate" and SO43- inconsistent and think most readers would prefer the use of one or the other rather than bouncing back and forth.

*We have changed all occurrences of the word sulfate to $SO_4^{2-}$, except with it was initially used (in same paragraph) and when used at the beginning of a sentence*

Sentence beginning L102: needs a grammatical revision for clarity. If "there" is referring to the forests, it should be "their" – and the entire sentence is difficult to follow.

*We have clarified the ending of this sentence. Including correcting the grammatical error.*

L111: run-on sentence

*We have added a semicolon after the citation.*

L131: needs a comma after "dynamics" (run-on sentence)

*We have added a comma*

L912 surrounding states water bodies – should that be "states' "?

*This sentence was awkwardly worded with that, so we have removed the word "states"*

L190: SLR rather than "sea level rise"

*We have changed this here and in a couple other places.*

L215: What is "fresh" soil?

*These are samples stored at 4C after pre processing and until the start of the incubation. We have added a note stating what this term refers to.*

L258: grammar- use "were" not "was"

*We have corrected this.*

L307: assuming the numerator intends to be the difference between delC X cond of fumigated and non-fumigated? If so, additional parentheses are needed, else the order of operations will change the results

*The reviewer is correct. Thank you for catching that. We have added more parenthesis in the numerator*

L319+: I greatly appreciate the authors adding rationale for the chosen enzymes. Will all your readers know what the "EC a.b.c.d" references are for? A brief indication of Enzyme Commission numbers might be helpful?

*We have added a sentence describing what EC stands for and what the EC number is used for*

L479-480: keep one "is" that was stricken

*We have added one back*

L526: subject-verb agreement needs correction (e.g. appear, not appears)

*We have made this change*

L566: comma splice – remove the comma after "incubations"

*We have made this change*

L605: please provide data supporting this statement comparing productivity, as salt marshes generally considered one of the most productive systems on the planet

*Our initial thought was that forested wetlands likely contain a much larger amount of aboveground biomass compared to marshes but this appears to not necessarily be supported by the literature. Therefore, we agree with the reviewer and have removed this part of the sentence.*

[revised manuscript text omitted]

---

## Author Response (AR3)

Helge,

We have addressed the final comments by you. We appreciate the timely and in depth review by you and the other two reviewers.

Kevan

**Response to Comments:**

Check sulfate vs SO42-

> *The manuscript has been checked for consistency. Sulfate is written out when it is first defined as SO42- and then at two other instances when it is the first word of the sentence.*

Fig 1 needs a brush-up, the overview map contains very small print, and the dash-dot lines are not clear. I would perhaps use a map of the entire East Coast of Canada/US rather than N-Carolina for the overview.

> We have added a map of North America, increased font size of print, and clarified in the figure caption what the dotted arrows indicate

Fig 2. Use the same font style for all items in the figure (some with, some without serif). The small letters are not really conclusive (statistically different to what?). Perhaps there is another way to indicate that?

> The letters do make the figures busy and aren't very clear. Therefore, we have removed the lettering asterisks over each time period that have significant differences. The differences by treatment are discussed in the text. Also, to try and improve the figures visually we have removed the lines between symbols and reduced the size of the symbol slightly.

> In the figure captions we had stated that "Treatment means with different lowercase letters are significantly different within a sampling time point ($P < 0.05$)." We have changed this to "An asterisk ($P < 0.05$) indicates significant differences between treatment means at each time point." to hopefully also make it more clear.

> The fonts appear to be the same for elements of the figure.

Fig 4. See comment to fig 2 for small letters

> Revised in same way as Figure 2.